# Adaptive Collaboration with Humans: Metacognitive Policy Optimization for Multi-Agent LLMs with Continual Learning

**Wei Yang,**\* **Defu Cao,**\* **Jiacheng Pang, Muyan Weng, Yan Liu**
University of Southern California
{wyang930,defucao,pangj,muyanwen,yanliu.cs}@usc.edu

## Abstract

While scaling individual Large Language Models (LLMs) has delivered remarkable progress, the next frontier lies in scaling collaboration through multi-agent systems (MAS). However, purely autonomous MAS remain "closed-world" systems, constrained by the static knowledge horizon of pre-trained models. This limitation makes them brittle on tasks requiring knowledge beyond training data, often leading to collective failure under novel challenges. To address this, we propose the **Human-In-the-Loop Multi-Agent Collaboration (HILA)** framework, a principled paradigm for human–agent collaboration. HILA trains agents to learn a metacognitive policy that governs when to solve problems autonomously and when to defer to a human expert. To operationalize this policy, we introduce **Dual-Loop Policy Optimization**, which disentangles immediate decision-making from long-term capability growth. The inner loop applies Group Relative Policy Optimization (GRPO) with a cost-aware reward to optimize deferral decisions, while the outer loop implements continual learning, transforming expert feedback into high-quality supervised signals that strengthen the agent's reasoning ability. Experiments on challenging mathematical and problem-solving benchmarks show that HILA, equipped with Dual-Loop Policy Optimization, consistently outperforms advanced MAS, establishing a principled foundation for collaborative and continually improving agentic systems. The code is available at https://github.com/USC-Melady/HILA.git.

## 1 Introduction

While scaling individual Large Language Models (LLMs) has produced remarkable progress, the next frontier lies in scaling collaboration through *multi-agent systems* (MAS) (Hong et al., 2023; Chen et al., 2023b; Jiang et al., 2023; Ning et al., 2023; Han et al., 2025; Wang et al., 2025a; Chen et al., 2025b). By coordinating multiple agents to tackle problems beyond the reach of any single model, this paradigm has inspired a wave of innovations from structured debates to dynamic workflow optimization (Zhang et al., 2024a; Qiao et al., 2024; Han et al., 2025). Yet these systems face an inherent ceiling: no matter how sophisticated their interaction protocols, purely autonomous agents remain fundamentally **closed-world**. Their knowledge horizon is bounded by pre-training corpora (Wang et al., 2023b; Srivatsa et al., 2024; Du et al., 2023; Liu et al., 2024). While they can recombine existing information, they cannot generate new knowledge or adapt to unseen contexts. This creates vulnerabilities when tasks demand real-time information, domain-specific expertise, or reasoning patterns absent from training (Zhang et al., 2024d; Chen et al., 2025a). In such cases, internal collaboration alone cannot bridge the gap, often leading to collective failure. To break this ceiling and enable open-ended intelligence, a new paradigm is needed. We argue that the most principled path is to integrate **external human expertise**, transforming closed systems into adaptive frameworks capable of continual learning and growth (Sun et al., 2025; Zou et al.).

Within this closed-world paradigm, research has followed two main directions. The first emphasizes optimizing **autonomous collaboration** through increasingly sophisticated interaction proto-

---
\*Equal contribution.

cols. Frameworks based on structured debate (Chan et al., 2023; Liu et al., 2024), topology control (Ong et al., 2024; Chen et al., 2024b), and workflow graph optimization (Zhang et al., 2024b; Li et al., 2025a) have demonstrated notable improvements in refining and recombining agents' internal knowledge. However, these methods largely engage in *collective introspection* (Zhang et al., 2024e; Chen et al., 2024a), maximizing the use of existing information without extending beyond the aggregate knowledge boundary. They act as powerful integrators, but not true learners capable of acquiring genuinely new capabilities. Recognizing this intrinsic limitation, a second line of work has sought to incorporate **human expertise** (Takerngsaksiri et al., 2025; Mozannar et al., 2025). Many human-in-the-loop systems (Liu et al., 2023; Pandya et al., 2024) treat humans primarily as passive oracles or supervisors for sub-tasks. This leaves two critical questions unresolved: *when* to defer to the expert, often reduced to heuristics such as low-confidence thresholds rather than learned policies (Kenton et al., 2024; Li et al., 2024b); and *how* to learn from human input, which is typically applied as a one-time fix rather than as a catalyst for long-term capability growth (Mu et al., 2024; Wang et al., 2025b). Importantly, human intervention holds the potential to operate at multiple levels, offering both localized corrections to specific reasoning errors and broader adjustments that reshape the overall collaborative process (Triem & Ding, 2024; Grondin et al., 2025).

This analysis highlights that the key challenge is not whether agents can interact with humans, but whether they can do so intelligently and strategically. Addressing this requires a **metacognitive policy**, a high-level strategy for reasoning about both self-competence and peer competence to guide collaboration. Such a policy must solve two intertwined problems: **when to ask**, which demands moving beyond heuristics to model uncertainty and balance the risk of failure against the cost of intervention; and **how to grow**, which requires mechanisms that turn expert feedback into lasting capability improvements rather than one-time fixes. A paradigm that unifies these elements is essential for building open and continually evolving agentic systems.

To address these challenges, we propose **Human-In-the-Loop Multi-Agent Collaboration (HILA)**, a principled framework for adaptive human–agent collaboration. The central idea of HILA is not simply to place a human in the loop, but to equip agents with a metacognitive policy that decides when external expertise is needed and how it should be used. HILA instantiates this idea through three coordinated components: (*i*) **Autonomous Operation**, where agents first attempt to solve problems using their current capabilities; (*ii*) **Metacognitive Assessment**, where they assess their own confidence and the difficulty of the task to recognize potential knowledge boundaries; and (*iii*) **Strategic Deferral**, where external expertise is invoked as a targeted intervention rather than a passive fallback. To optimize this metacognitive behavior, we further introduce **Dual-Loop Policy Optimization (DLPO)**, a training framework that separates short-term intervention decisions from long-term capability improvement. Its inner loop applies Group Relative Policy Optimization (GRPO) with a cost-aware reward to refine deferral behavior online, while its outer loop turns expert feedback from deferral events into supervised training signals that continually strengthen the model's underlying reasoning ability.

In summary, the main contributions of this paper are as follows:

- We propose the **Human-in-the-Loop Multi-Agent Collaboration (HILA)** framework, a paradigm for human–agent collaboration that equips agents with a metacognitive policy to decide when to strategically defer to human expertise.

- We introduce **Dual-Loop Policy Optimization (DLPO)**, a training methodology that separates short-term deferral decisions from long-term capability growth. The inner loop employs GRPO with a cost-aware reward, while the outer loop leverages expert feedback as supervised signals for continual learning.

- Extensive experiments on mathematical reasoning and general problem-solving benchmarks demonstrate that HILA with DLPO outperforms both autonomous multi-agent systems, establishing a robust foundation for continually improving agentic collaboration.

## 2 RELATED WORK

Large language models (LLMs) acting alone are limited by context length, sequential generation, and restricted skill coverage, which constrains their ability to solve complex reasoning tasks (Gabriel et al., 2024; Ping et al., 2025b; Li et al., 2025b; Ye et al., 2024; Xia et al., 2025a). To mitigate these

issues, **multi-agent systems** (MAS) have been widely explored, where multiple LLMs are organized into collaborative structures for collective problem solving (Hong et al., 2023; Yang et al., 2025a; Jiang et al., 2023; Qiao et al., 2024; Yang & Thomason, 2025). Early efforts relied on prompt-based paradigms that assign predefined roles or workflows, enabling debate, critique, or corporate-style pipelines (Du et al., 2023; Chan et al., 2023; Yang et al., 2025b; Chang et al., 2025). While effective, these designs lack adaptability since their interaction protocols are fixed and cannot evolve through experience. More recent work moves toward structured coordination and adaptive communication. Predefined schemes employ debate or peer-review across chains, trees, or graphs (Liu et al., 2024; Qian et al., 2024), while adaptive methods restructure interactions dynamically via routing, pruning, or workflow search (Yang et al., 2026; Chen et al., 2026; Yue et al., 2025; Ping et al., 2025a).

A complementary line of research introduces **human-in-the-loop** collaboration. Humans have been positioned as supervisors, oracles, or evaluators, providing corrections or domain knowledge to strengthen agent performance (Takerngsaksiri et al., 2025; Mozannar et al., 2025; Liu et al., 2023; Pandya et al., 2024). Closely related, Siedler & Gemp (2025) study LLM-mediated guidance in MARL, where an LLM serves as a natural-language controller that interprets and delivers interventions to shape agents' learning trajectories and accelerate training. However, such systems often rely on heuristics (e.g., confidence thresholds) to trigger deferral (Kenton et al., 2024; Li et al., 2024b), and feedback is usually treated as a one-time fix rather than a signal for sustained capability growth (Mu et al., 2024; Wang et al., 2025b). Recent discussions highlight that human involvement can occur at multiple levels, from correcting local reasoning errors to reshaping global collaborative dynamics (Triem & Ding, 2024; Grondin et al., 2025). Together, these directions have advanced the field of MAS, yet challenges remain in moving beyond closed-world recombination toward open and adaptive collaboration. A detailed review of related work is provided in Appendix A.

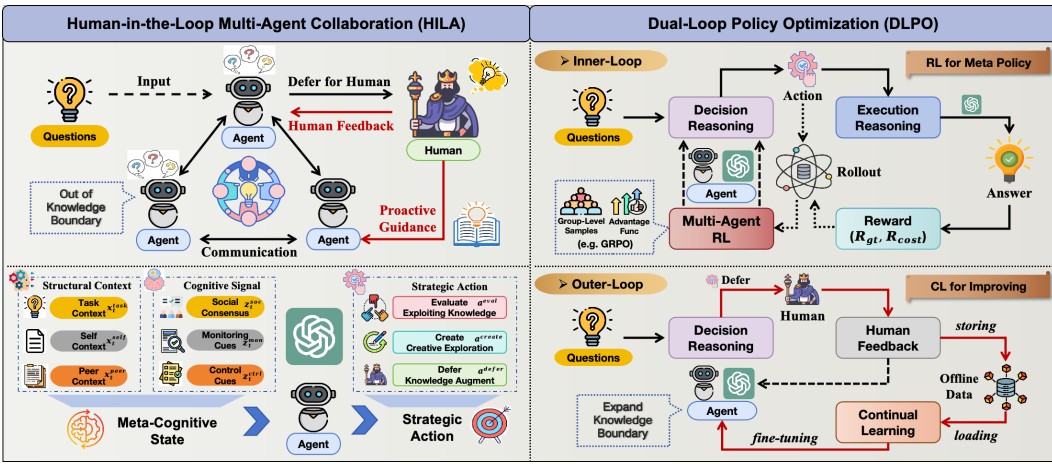

Figure 1: Overview of the proposed HILA framework and its Dual-Loop Policy Optimization (DLPO) training paradigm. Left: HILA coordinates multi-agent collaboration with both proactive human guidance and reactive expert feedback via metacognitive states and strategic actions (EVAL, CREATE, DEFER). Right: DLPO optimizes the meta-policy in an inner RL loop with cost-aware rewards, and expands the model's knowledge boundary in an outer continual-learning loop by storing DEFER-triggered human feedback as offline supervision.

## 3 METHODOLOGY

Our methodology builds a multi-agent system designed for adaptive collaboration with a human expert. Figure 1 provides an overview of HILA and its Dual-Loop Policy Optimization, highlighting how multi-agent collaboration is coupled with human. At its core is a **metacognitive policy** that enables agents to reason about both their own competence and that of their peers, thereby deciding when to act autonomously and when to defer to external expertise. We formalize this collaborative process as a Metacognitive Markov Decision Process, which provides the foundation for our framework. The framework is then specified through a structured cognitive state space and a functional action space. Finally, we introduce a Dual-Loop Policy Optimization algorithm that combines reinforcement learning to refine the metacognitive policy with continual learning to integrate expert feedback into lasting capability growth.

### 3.1 PRELIMINARIES: THE METACOGNITIVE MARKOV DECISION PROCESS

We model human–agent collaboration as a **Metacognitive Markov Decision Process (Meta-MDP)**, which formalizes decision-making over high-level cognitive strategies such as autonomous problem-solving or deferral to human expertise. This abstraction provides a principled foundation for defining states, actions, transitions, and rewards in our collaborative framework. The full formalization and detailed design are provided in Appendix B.

### 3.2 A FRAMEWORK FOR HUMAN-AGENT COLLABORATION

Building on the Meta-MDP, we introduce a framework that operationalizes human–agent interaction through three components: (i) a structured cognitive state space that encodes problem context and metacognitive assessments, (ii) a functional action space representing high-level collaboration strategies, and (iii) an interaction protocol that specifies coordination across rounds. These elements allow agents to reason about tasks while regulating their decision boundaries in a principled way.

#### 3.2.1 STRUCTURED COGNITIVE STATE SPACE

A metacognitive policy should not make decisions based solely on a narrow local response, since effective coordination depends on a broader view of the current reasoning context. In our setting, the policy should consider three main sources of information. First, it needs access to the **task context**, including the original question and relevant interaction history, which defines the decision environment and the current objective. Second, it should observe the agent's own **self context**, namely its latest solution and local reasoning status, which reflect its current belief and confidence. Third, it should incorporate **peer context** from other agents, since their responses provide complementary evidence about agreement, disagreement, and alternative solution paths. Together, these sources provide a richer basis for deciding whether to preserve the current trajectory, generate a new solution, or seek stronger external assistance.

To further support meta-policy decisions beyond the raw task/self/peer context, we optionally augment $s_t$ with a compact set of **structured cognitive signals** that instantiate a monitoring–control view of metacognition together with standard social-information cues. Under uncertainty, the controller benefits from three complementary evidence sources: *social evidence* about the collective state Cannon-Bowers et al. (1993); Jost et al. (2013), *self-monitoring evidence* about the reliability of the current solution Nelson & Narens (1994); Dunlosky & Metcalfe (2008), and *cognitive control evidence* about whether further internal deliberation is likely to help or whether escalation is warranted Risko & Gilbert (2016). We operationalize these sources via lightweight summaries over observable interaction traces: **social consensus cues** $\mathbf{z}_t^{\text{soc}}$ (convergence vs. conflict), **metacognitive monitoring cues** $\mathbf{z}_t^{\text{mon}}$ (local reliability), and **cognitive control cues** $\mathbf{z}_t^{\text{ctrl}}$ (progress vs. escalation). All cues are computed with simple parsing- and rule-based heuristics, yielding an explicit decision-oriented abstraction without introducing additional learned modules.

Formally, we represent the policy state as

$$s_t = \text{concat}\big(\mathbf{x}_t^{\text{task}}, \mathbf{x}_t^{\text{self}}, \mathbf{x}_t^{\text{peer}}, \mathbf{z}_t^{\text{soc}}, \mathbf{z}_t^{\text{mon}}, \mathbf{z}_t^{\text{ctrl}}\big), \tag{1}$$

where $\mathbf{x}_t^{\text{task}}$, $\mathbf{x}_t^{\text{self}}$, and $\mathbf{x}_t^{\text{peer}}$ denote the task, self, and peer context, respectively, and $\mathbf{z}_t^{\text{soc}}$, $\mathbf{z}_t^{\text{mon}}$, and $\mathbf{z}_t^{\text{ctrl}}$ are optional structured cognitive cues capturing social consensus, metacognitive monitoring, and metacognitive control evidence. In practice, these cues are computed via lightweight parsing- and rule-based heuristics over the observable interaction traces, rather than introducing additional learned modules or external supervision. This design provides the meta-policy with an explicit, decision-oriented abstraction of the current collaborative state while keeping the structured component lightweight and auxiliary.

#### 3.2.2 THE STRATEGIC ACTION SPACE

The action space $\mathcal{A}$ in our Meta-MDP is not defined by low-level text generation, but by a discrete set of high-level cognitive strategies. These actions empower the agent to manage its problem-solving process, balancing the exploitation of existing collective knowledge against the exploration of new solutions and the strategic deferral to external expertise. Formally, the action space is defined as $\mathcal{A} = \{a^{\text{eval}}, a^{\text{create}}, a^{\text{defer}}\}$.

**Evaluate ($a^{\text{eval}}$): Exploiting Collective Knowledge.** This action represents the cognitive stance of convergence and synthesis. When selecting $a^{\text{eval}}$, the agent commits to exploiting the existing knowledge within the multi-agent group. Operationally, it must select and endorse one of the solutions already proposed by its peers in the current round. This action allows the agent to leverage collective intelligence and reinforce high-quality, consensual solutions.

**Create ($a^{\text{create}}$): Creative Exploration and Hypothesis Generation.** This action embodies the cognitive stance of divergence and exploration. By choosing $a^{\text{create}}$, the agent posits that the current solution pool is insufficient and commits to generating a novel solution sequence ('Choice', 'Reason') from scratch. This action is crucial for breaking cognitive fixation, correcting shared errors within the group, and introducing new, potentially superior reasoning paths.

**Defer ($a^{\text{defer}}$): Risk Mitigation and Knowledge Augmentation.** This action represents the highest level of metacognitive awareness—the ability to recognize the limits of the system's own capabilities. Selecting $a^{\text{defer}}$ signals that the agent assesses the problem's uncertainty or difficulty to be beyond the collective's current ability to solve reliably. Operationally, this triggers a call to the external human expert, whose high-quality demonstration is then used as the round's output. This action serves as both a mechanism for ensuring task success in critical situations and as a conduit for introducing new knowledge into the system via continual learning.

### 3.2.3 COLLABORATIVE INTERACTION MODEL

A defining feature of our framework is the integration of human expertise into collective reasoning through a structured, multi-round protocol. At each round $t$, all $N$ agents receive the shared cognitive state $s_t$. Each agent $i$ independently samples a metacognitive action $a_{i,t} \sim \pi_\theta(a|s_t)$ and executes it in parallel. The output $y_{i,t}$ depends on the chosen action. When acting autonomously, the agent applies its internal generation process $g_\theta(s_t)$. When deferring, it adopts the authoritative solution $y_{\text{human},t}$ provided by the expert:

$$y_{i,t} = \begin{cases} g_\theta(s_t), & \text{if } a_{i,t} \in \{a^{\text{eval}}, a^{\text{create}}\}, \\ y_{\text{human},t}, & \text{if } a_{i,t} = a^{\text{defer}}. \end{cases} \tag{2}$$

The collection $\{y_{1,t}, \ldots, y_{N,t}\}$ then forms the next state $s_{t+1}$, ensuring that updates reflect the most reliable signals, whether from autonomous synthesis or human demonstration.

From a learning perspective, the *Defer* action plays a dual role. It serves as risk mitigation, ensuring progress under uncertainty by overriding flawed solutions, and as knowledge augmentation, injecting expert demonstrations as high-quality samples for continual learning (Section 3.3). Thus, the human collaborator functions not merely as a fallback oracle but as a driver of system improvement.

### 3.3 ADAPTIVE POLICY OPTIMIZATION WITH CONTINUAL LEARNING

Mastering the metacognitive challenges described above requires an optimization strategy that balances two competing paths: the high-risk but potentially high-reward route of autonomous problem-solving, and the low-risk but constrained option of deferring to an expert. This trade-off naturally lends itself to reinforcement learning (RL), where the goal is to learn a policy $\pi_\theta$ that maximizes the expected utility of the collaborative process. To address this, we propose a **Dual-Loop Policy Optimization (DLPO)** framework that integrates a reinforcement learning objective for strategic policy optimization with a supervised objective for continual knowledge acquisition.

### 3.3.1 INNER LOOP: REINFORCEMENT LEARNING FOR METACOGNITIVE POLICY

The inner loop optimizes the agent's high-level policy $\pi_\theta(a|s_t)$ over strategic actions. The key challenge is to provide a learning signal that reflects the trade-off between autonomous success, potential failure, and the cost of expert intervention. This problem is well-suited to **Group Relative Policy Optimization (GRPO)**, which contrasts the relative advantages of actions in each state.

**Reward Formulation.** We use a reward function $R(s_t, a_t)$ that combines task correctness with lightweight action-dependent costs. Let $\hat{y}(a_t)$ denote the output obtained after executing action $a_t$

at state $s_t$, and let $R_{\text{gt}}(\hat{y})$ denote the task reward induced by the resulting answer (e.g., a binary correctness signal). We define

$$R(s_t, a_t) = \begin{cases} R_{\text{gt}}(\hat{y}(a_t)), & a_t = \texttt{EVAL}, \\ R_{\text{gt}}(\hat{y}(a_t)) - C_{\text{create}}, & a_t = \texttt{CREATE}, \\ R_{\text{gt}}(\hat{y}_{human}(a_t)) - C_{\text{defer}}, & a_t = \texttt{DEFER}, \end{cases} \tag{3}$$

where $C_{\text{create}}$ and $C_{\text{defer}}$ are small tunable penalties reflecting the additional cost of internal regeneration and external expert intervention, respectively, with $C_{\text{defer}} > C_{\text{create}} \geq 0$. In this way, correctness remains the primary training signal, while the cost terms encourage the policy to prefer lower-cost actions when multiple choices lead to similarly good outcomes.

**GRPO Objective.** Given the reward vector $\mathbf{R}_t = [R(s_t, a_1), \ldots, R(s_t, a_K)]$, advantages are computed by centering rewards:

$$A(s_t, a_k) = R(s_t, a_k) - \frac{1}{K} \sum_{j=1}^{K} R(s_t, a_j). \tag{4}$$

The policy gradient objective is:

$$\mathcal{L}_{\text{PG}}(\theta) = -\mathbb{E}_{s_t, a_t \sim \pi_\theta} \left[ A(s_t, a_t) \log \pi_\theta(a_t | s_t) \right]. \tag{5}$$

Two regularizers ensure stability: a KL-penalty constrains deviation from the reference policy $\pi_{\text{ref}}$, and an entropy bonus promotes exploration. The final inner-loop loss is:

$$\mathcal{L}_{\text{Inner}} = \mathcal{L}_{\text{PG}} + \beta_{\text{kl}} \mathcal{L}_{\text{KL}} - \beta_{\text{ent}} \mathcal{L}_{\text{Entropy}}. \tag{6}$$

### 3.3.2 OUTER LOOP: CONTINUAL LEARNING FROM EXPERT FEEDBACK

While the inner loop optimizes how the agent *uses* its current abilities, a truly adaptive system must also *expand* them. Reinforcement learning alone cannot overcome the knowledge ceiling of the base LLM, as it improves decision policies without introducing fundamentally new skills. To break this ceiling, we introduce an outer optimization loop for **Continual Learning from Expert Demonstrations**.

This loop is activated by the 'Defer' action, which indicates that the agent has identified a knowledge gap. When deferring, the agent receives a high-quality demonstration $y_{\text{human}} = (t_1, \ldots, t_L)$ from the expert, which is converted into a supervised fine-tuning (SFT) sample. The training objective is to maximize the likelihood of this sequence, conditioned on the state $s_t$, by minimizing the cross-entropy loss:

$$\mathcal{L}_{\text{SFT}}(\theta) = -\sum_{i=1}^{L} \log \pi_\theta(t_i \mid s_t, t_{1:i-1}). \tag{7}$$

In this design, the inner RL loop determines *when* to defer, while the outer loop teaches *what* to learn from expert input. Together, they establish an apprentice–mentor dynamic: the agent strategically invokes human guidance and systematically assimilates it into lasting capability growth.

### 3.3.3 THE FINAL DUAL-LOOP POLICY OPTIMIZATION OBJECTIVE

The inner and outer loops are optimized jointly to train a single agent that is both strategically adept and continually improving. The final training objective is a principled combination of the reinforcement learning signal from the inner loop and the conditional supervised signal from the outer loop. The total loss, $\mathcal{L}_{\text{total}}$, is computed over a batch of experiences:

$$\mathcal{L}_{\text{total}}(\theta) = \mathbb{E}_{(s_t, a_t)} \left[ \mathcal{L}_{\text{Inner}}(\theta) + \lambda_{\text{sft}} \cdot \mathbb{I}(a_t = a^{\text{defer}}) \cdot \mathcal{L}_{\text{SFT}}(\theta) \right], \tag{8}$$

where $\mathcal{L}_{\text{Inner}}(\theta)$ is the full GRPO objective, $\lambda_{\text{sft}}$ is a hyperparameter balancing the two learning signals, and $\mathbb{I}(\cdot)$ is the indicator function that ensures the SFT loss is only applied when the 'Defer' action is taken.

| Model | Type | GSM8K | AMC | AIME | HumanEval | MMLU |
|-------|------|-------|-----|------|-----------|------|
| Vanilla | SA | 72.76 (+0.00) | 8.03 (+0.00) | 2.96 (+0.00) | 47.56 (+0.00) | 57.99 (+0.00) |
| CoT | SA | 74.22 (+1.46) | 11.65 (+3.62) | 3.70 (+0.74) | 51.42 (+3.86) | 61.57 (+3.58) |
| SC | SA | 80.79 (+8.03) | 12.45 (+4.42) | 4.07 (+1.11) | 57.52 (+9.96) | 68.30 (+10.31) |
| PHP | MA | 80.01 (+7.25) | 15.66 (+7.63) | 4.44 (+1.48) | 56.50 (+8.94) | 68.46 (+10.47) |
| Debate | MA | 83.52 (+10.76) | 19.28 (+11.25) | 5.56 (+2.60) | 57.72 (+10.16) | 67.59 (+9.60) |
| G-Debate | MA | 83.98 (+11.22) | 20.48 (+12.45) | 5.19 (+2.23) | 57.93 (+10.37) | 69.89 (+11.90) |
| DyLAN | MA | 82.03 (+9.27) | 19.68 (+11.65) | 3.70 (+0.74) | 61.59 (+14.03) | 66.85 (+8.86) |
| G-Swarm | MA | 84.89 (+12.13) | 15.66 (+7.63) | 5.78 (+2.82) | 59.55 (+11.99) | 69.67 (+11.68) |
| A-Prune | MA | 84.38 (+11.62) | 16.47 (+8.44) | 4.81 (+1.85) | 57.11 (+9.55) | 69.09 (+11.10) |
| AFlow | MA | 83.75 (+10.99) | 12.05 (+4.02) | 4.44 (+1.48) | 62.20 (+14.64) | 69.31 (+11.32) |
| HILA | MA | **89.86** (+17.10) | **35.83** (+24.47) | **9.37** (+6.41) | **72.15** (+24.59) | **73.62** (+15.63) |

Table 1: Comparison of baseline and proposed methods using the LLaMA3-8B backbone. All values are percentages (the percent sign is omitted in the table). Values in parentheses denote absolute differences relative to the *Vanilla* baseline (first row). Underlined numbers indicate the best-performing baseline on each benchmark. "SA" denotes single-agent, and "MA" denotes multi-agent settings.

## 4 EXPERIMENTS

**Experimental Setup.** We evaluate our method on a broad suite of benchmarks, including general language understanding (*MMLU*), program synthesis (*HumanEval*), and quantitative mathematics (*GSM8K*, *AIME*, *AMC*). Following related works (Liu et al., 2023; Pandya et al., 2024), we employ **GPT-4o-mini** as a proxy human expert, leveraging its strong reasoning capability to simulate human interventions. Detailed experimental and training settings are provided in Appendix C.1.

**Overall Performance.** Table 1 shows that our HILA framework with DLPO achieves the best overall performance among all compared methods, consistently surpassing strong autonomous multi-agent baselines across all reported benchmarks. These baselines, including debate-style (*e.g.*, LLM-Debate), topology-based (*e.g.*, DyLAN), and graph-optimization (*e.g.*, GPTSwarm, AFLOW) methods, remain confined to "closed-world" collaboration, where performance is capped by the agents' internal knowledge and often degrades on problems requiring non-obvious reasoning paths. In contrast, HILA introduces an "open-world" dynamic by enabling agents to strategically access external expertise, directly addressing this knowledge ceiling. On the Llama3-8B backbone, HILA yields substantial gains over the strongest autonomous baseline on every benchmark, with absolute improvements ranging from 3.7 to 15.4 points. The gains are particularly pronounced on competition-style math benchmarks such as AMC and AIME, where cascade failures from flawed premises are common. By learning a metacognitive policy to defer under high uncertainty, HILA avoids these pitfalls and effectively leverages superior guidance. These results suggest that the performance gains arise not merely from more complex interaction, but from the principled integration of external knowledge and the agent's learned ability to decide when to invoke it.

**Cross-Backbone Generalization.** We further evaluate HILA across four heterogeneous LLM backbones, spanning both the Qwen and LLaMA families and covering multiple model scales. As shown in Table 2, HILA consistently achieves the best performance on GSM8K across all backbones, outperforming both single-agent prompting baselines and strong autonomous multi-agent methods. This result shows that the effectiveness of HILA is not tied to a specific model family or parameter scale, but transfers robustly across substantially different pretrained backbones. Moreover, the gains are especially pronounced on smaller or weaker models, suggesting that HILA can more effectively compensate for limited intrinsic reasoning capacity when the base model is less capable. Taken together, these results demonstrate that the proposed framework is both general across backbone choices and scalable across model strengths, while maintaining consistent benefits from adaptive collaboration and strategic human intervention.

**Beyond Better Intervention Decisions.** To further analyze the source of HILA's performance, we separate the contributions of *policy learning* and *capability growth*. Table 3 compares three variants of HILA under progressively stronger training regimes. Moving from the unoptimized policy to

| Model | Qwen2.5-7B | Qwen2.5-3B | LLaMA3-8B | LLaMA3-3B |
|---|---|---|---|---|
| Vanilla | 90.71 (+0.00) | 83.25 (+0.00) | 72.76 (+0.00) | 45.26 (+0.00) |
| CoT | 91.13 (+0.42) | 84.36 (+1.11) | 74.22 (+1.46) | 52.49 (+7.23) |
| SC | 91.79 (+1.08) | 87.54 (+4.29) | 80.79 (+8.03) | 58.68 (+13.42) |
| PHP | 91.35 (+0.64) | 86.83 (+3.58) | 80.01 (+7.25) | 66.21 (+20.95) |
| LLM-Debate | 92.47 (+1.76) | 87.41 (+4.16) | 83.52 (+10.76) | 76.32 (+31.06) |
| DyLAN | 92.64 (+1.93) | 87.85 (+4.60) | 82.03 (+9.27) | 76.17 (+30.91) |
| GPTSwarm | 92.83 (+2.12) | 86.45 (+3.20) | 84.89 (+12.13) | 71.85 (+26.59) |
| AgentPrune | 91.76 (+1.05) | 86.20 (+2.95) | 84.38 (+11.62) | 68.71 (+23.45) |
| AFlow | 92.37 (+1.66) | 86.78 (+3.53) | 83.75 (+10.99) | 70.89 (+25.63) |
| HILA | **94.72** (+4.01) | **91.17** (+7.92) | **89.86** (+17.10) | **83.85** (+38.59) |

Table 2: Performance of baselines across four LLM backbones on GSM8K. All values are percentages (percent sign omitted). Parentheses show absolute differences (percentage points) relative to the *Vanilla* row for each backbone. HILA refers to the proposed method with DLPO.

| Model | GSM8K | AMC | MMLU |
|---|---|---|---|
| HILA (Init Policy) | 88.15 | 33.33 | 68.30 |
| HILA + GRPO | 88.38 | 32.50 | 70.47 |
| HILA + DLPO | **89.86** | **35.83** | **73.62** |

Table 3: Performance of HILA under progressively stronger training regimes. *Init Policy* denotes the unoptimized system, *GRPO* applies only the inner-loop policy optimization, and *DLPO* adds the outer-loop update from expert demonstrations. The results show that the full dual-loop objective delivers the strongest overall performance.

| Model | GSM8K | AMC | MMLU |
|---|---|---|---|
| Vanilla (Base) | 72.76 | 8.03 | 57.99 |
| Vanilla (DLPO) | **82.11** | **10.84** | **61.58** |
| DyLAN (Base) | 82.03 | 19.68 | 66.85 |
| DyLAN (DLPO) | **88.32** | **25.30** | **68.72** |
| Debate (Base) | 83.52 | 19.28 | 67.59 |
| Debate (DLPO) | **88.93** | **21.69** | **69.57** |

Table 4: Transferability of the backbone updated by DLPO. *Base* denotes the original LLaMA3-8B backbone, while *DLPO* denotes the same backbone after DLPO training. For each inference framework, we keep the test-time protocol unchanged and only replace the backbone. All values are percentages.

GRPO yields only modest changes overall, with the main benefit appearing in more reliable strategic control rather than uniformly higher task accuracy. In contrast, introducing the full DLPO objective leads to a clear second-stage improvement across benchmarks, indicating that the gains cannot be explained by better action selection alone. This pattern suggests that reinforcement learning primarily improves *when* the system chooses to defer, while the outer supervised loop is responsible for transforming those deferral events into lasting gains in reasoning ability.

Table 4 further verifies this interpretation by evaluating standard inference pipelines with the backbone updated after DLPO training. Across single-agent prompting and autonomous multi-agent baselines, replacing the original backbone with the DLPO-updated one consistently improves performance, showing that the benefit transfers beyond the HILA interaction protocol itself. Notably, these gains appear even when the downstream method does not use strategic deferral at inference time, which indicates that the expert demonstrations collected during training are not merely helping HILA make better local decisions, but are also improving the underlying LLM's general reasoning competence. Taken together, these results show that HILA's gains arise from two complementary sources: a better metacognitive policy for intervention, and continual learning that strengthens the backbone model itself.

**Policy Distribution Across Training Stages.** To better understand how training changes HILA's collaboration behavior, we examine the distribution of the three strategic actions (EVAL, CREATE, and DEFER) across different training stages. Table 5 shows a clear and interpretable trend. Starting from the unoptimized policy, HILA assigns a nontrivial fraction of decisions to DEFER, indicating substantial reliance on external intervention before the system learns a calibrated metacognitive strategy. After applying GRPO, the share of DEFER decreases consistently across datasets, while EVAL and CREATE become slightly more frequent. This trend suggests that inner-loop policy opti-

| Model | GSM8K | | | AMC | | | MMLU | | |
|---|---|---|---|---|---|---|---|---|---|
| | EVAL | CREATE | DEFER | EVAL | CREATE | DEFER | EVAL | CREATE | DEFER |
| HILA (Init Policy) | 36% | 35% | 29% | 36% | 40% | 24% | 47% | 34% | 19% |
| HILA + GRPO | 38% | 36% | 26% | 40% | 40% | 20% | 48% | 37% | 15% |
| HILA + DLPO | 55% | 28% | 17% | 64% | 24% | 12% | 74% | 21% | 5% |

Table 5: Distribution of strategic actions across training stages. As training progresses, the proportion of DEFER consistently decreases, while EVAL becomes more dominant, indicating improved metacognitive control and reduced reliance on external intervention.

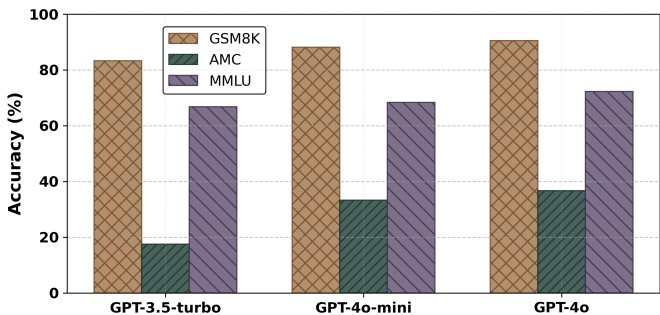

Figure 2: Effect of human proxy capability on HILA. Using stronger language models as the external expert consistently improves performance on GSM8K, AMC, and MMLU.

mization learns a more cost-aware intervention strategy. Because deferral carries an explicit penalty, the agent becomes more selective about invoking external expertise and makes better use of autonomous alternatives.

The effect becomes even more pronounced after full DLPO training. Compared with both the initialization stage and GRPO alone, DEFER drops substantially on all benchmarks, accompanied by a marked increase in EVAL. This shift suggests that the agent is not merely learning to avoid costly deferral heuristically, but is becoming more capable of resolving tasks within the multi-agent system. Since DLPO combines inner-loop policy optimization with continual learning from expert demonstrations, the reduced deferral rate further suggests that these demonstrations improve the model's underlying reasoning ability. As the backbone becomes stronger, the system relies less on external help and more often exploits or refines internally available solutions. These results show that GRPO improves *when* to ask, while DLPO further improves *whether asking is needed at all*.

**Effect of Human Proxy Capability.** We further study how the strength of the external expert shapes HILA's effectiveness by instantiating the human proxy with three language models of different capabilities, namely `gpt-3.5-turbo`, `gpt-4o-mini`, and `gpt-4o`. As shown in Figure 2 and Table 6, stronger proxies consistently lead to better overall performance across all benchmarks. This ordering is stable across GSM8K, AMC, and MMLU: `gpt-3.5-turbo` yields the weakest results, `gpt-4o-mini` provides a clear improvement, and `gpt-4o` achieves the best performance. This trend is consistent with the role of DEFER in HILA, which is designed to bring in higher-quality reasoning precisely when the autonomous multi-agent system reaches its limit. The results therefore show that HILA depends not only on learning *when* to defer, but also on *who* the system defers to. Strategic intervention and expert quality are complementary: a stronger metacognitive policy can identify when outside help is needed, while the final gain still depends on the informativeness and reliability of the returned guidance.

**The Value of Collective Exploration.** To examine how the size of the collective affects both effectiveness and cost, we evaluate HILA with different numbers of autonomous agents on AMC and MMLU, as shown in Figure 7(a), with full results provided in Appendix C.2, Table 7 and Table 8. Across both benchmarks, increasing the number of agents generally improves accuracy, with the most noticeable gains appearing when moving from a very small collective to a moderate-sized

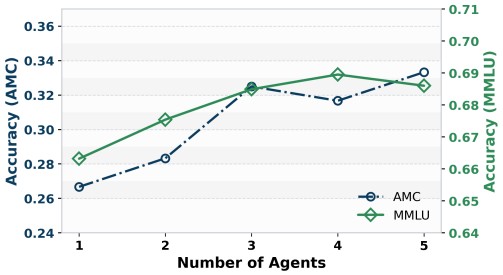 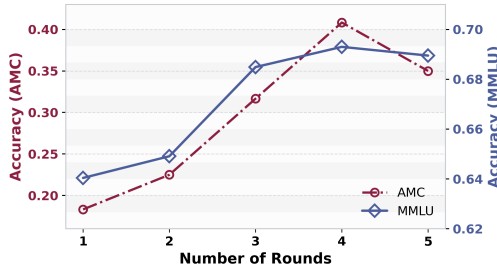

(a) Accuracy as a function of the number of agents.     (b) Accuracy as a function of the number of rounds.

Figure 3: Effect of scaling collaborative configurations. (a) shows how increasing the number of agents impacts accuracy on AMC and GSM8K, while (b) analyzes how varying the number of interaction rounds influences performance. Together, the results highlight the trade-offs between broader exploration through more agents and deeper refinement through additional rounds.

one. This trend supports a central intuition of HILA: a larger agent pool enables richer *collective exploration*, allowing the system to generate a broader set of candidate reasoning trajectories and increasing the chance that a strong solution can be identified and propagated. However, the gains are not linear. As the collective continues to grow, improvements become smaller and less stable, suggesting diminishing returns once the solution space is already sufficiently diversified. By contrast, computational cost rises sharply with the number of agents, as both input and output tokens increase substantially with broader participation. These results highlight a practical trade-off: larger collectives can strengthen reasoning through broader exploration, but this benefit must be balanced against rising inference cost, making the number of agents an important balance choice in HILA.

**Optimal Depth of Iterative Collaboration.** To assess the role of iterative refinement, we vary the number of collaborative rounds and evaluate HILA on AMC and MMLU, as shown in Figure 7(b), with full results provided in Appendix C.2, Table 9 and Table 10. Across both benchmarks, the effect follows a clear non-monotonic pattern. Increasing the number of rounds from a shallow interaction depth to a moderate one consistently improves accuracy, indicating that multi-round collaboration helps the system refine intermediate solutions, incorporate peer feedback, and correct earlier mistakes. This supports the intuition that iterative interaction strengthens reasoning by enabling agents to progressively revise and consolidate candidate solutions rather than relying on a single-pass exchange. However, the benefit does not continue indefinitely. Performance peaks at an intermediate depth and then declines slightly as the interaction is extended further, even though token usage continues to rise. This suggests that additional rounds initially create useful opportunities for correction, but excessive interaction leads to diminishing returns and can begin to reinforce suboptimal trajectories. These results show that iterative collaboration is valuable but not unbounded, and the number of rounds should be chosen to balance refinement gains against rising computational cost.

## 5   CONCLUSION

In this paper, we presented **HILA**, a framework that equips multi-agent systems with a metacognitive policy for deciding when to act autonomously and when to defer to external expertise. Through **Dual-Loop Policy Optimization**, which combines GRPO-based intervention control with continual learning from expert demonstrations, HILA enables both adaptive decision-making and long-term capability improvement. Extensive experiments across diverse and challenging reasoning benchmarks show that HILA consistently outperforms strong autonomous and multi-agent baselines. In future work, we plan to study more dynamic collaboration mechanisms and further strengthen the evolutionary capabilities of multi-agent systems.

ACKNOWLEDGEMENT

This work was supported in part by the Department of Defense under Cooperative Agreement Number W911NF-24-2-0133. The views and conclusions contained in this document are those of the authors and should not be interpreted as representing the official policies, either expressed or implied, of the Army Research Office or the U.S. Government. The U.S. Government is authorized to reproduce and distribute reprints for Government purposes notwithstanding any copyright notation herein.

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

# A RELATED WORK

## A.1 COLLABORATION PARADIGMS IN MULTI-AGENT LLM SYSTEMS.

Early research has shown that single LLM agents face inherent limitations in context length, sequential generation, and breadth of skills, which restrict their ability to solve complex tasks requiring diverse perspectives or parallel reasoning (Gabriel et al., 2024; Liang et al., 2023; Xiong et al., 2023; Yin et al., 2023; Zhang et al., 2023). To overcome these bottlenecks, recent work has explored **multi-agent systems** (MAS), where multiple LLMs are orchestrated to realize collective intelligence across domains such as software engineering, planning, and problem solving (Hong et al., 2023; Chen et al., 2023b; Xia et al., 2025b; Ning et al., 2023; Li et al., 2026; Pan et al., 2024; Suzgun & Kalai, 2024; Chen et al., 2023a; Weng et al., 2026).

Most existing frameworks rely on prompt-based paradigms that predefine roles, communication protocols, or workflow structures. These designs enable debate, critique, and corporate-style pipelines, achieving notable gains in coordination efficiency (Du et al., 2023; Chan et al., 2023; Cao et al., 2025; Mukobi et al., 2023; Wang et al., 2023a; Abdelnabi et al., 2024; Han et al., 2025). However, because they are hand-crafted and do not adapt through experience, such systems remain fundamentally **closed-world**: they can only recombine existing knowledge rather than acquire genuinely new capabilities (Wang et al., 2023b; Liu et al., 2024; Chen et al., 2024b).

Beyond fixed prompts, two further directions have emerged. **Prestructured coordination** employs fixed debate or peer-review topologies, such as chains, trees, or graphs, to refine reasoning (Du et al., 2023; Liu et al., 2024; Qian et al., 2024). In contrast, **self-organizing approaches** adapt the interaction graph dynamically through search, pruning, routing, or evolutionary mechanisms (Hu et al., 2024; Shang et al., 2024; Zhang et al., 2024b; Zhuge et al., 2024; Zhang et al., 2024c; Yue et al., 2025). These advances highlight the importance of who communicates and when, yet they primarily optimize internal coordination and leave unaddressed the problem of learning from external expertise.

In contrast, our work introduces a centralized and iterative collaboration framework that explicitly incorporates **human expertise** as an open-world resource. Rather than treating human feedback as a passive oracle or one-time correction (Takerngsaksiri et al., 2025; Mozannar et al., 2025; Liu et al., 2023; Pandya et al., 2024), we propose to endow agents with a metacognitive policy that governs both the timing of deferral and the assimilation of human guidance into lasting improvements. This approach differs fundamentally from prior debate, routing, and workflow-search systems, which often lack principled credit assignment between reasoning and decision outcomes (Chan et al., 2023; Talebirad & Nadiri, 2023; Wei et al., 2025). By integrating external knowledge with learned metacognitive adaptation, our framework moves beyond static collaboration to establish a pathway toward adaptive and continually improving multi-agent intelligence.

## A.2 MULTI-AGENT REINFORCEMENT LEARNING FOR LLMS.

A growing line of work seeks to move beyond static prompt engineering and endow multi-agent LLM systems with adaptive learning capabilities. Early efforts rely on supervised fine-tuning (SFT) to inject collaborative patterns by imitating expert demonstrations or curated trajectories (Lu et al., 2023; Madaan et al., 2023; Zelikman et al., 2022; Wei et al., 2021). While effective for seeding cooperative behaviors, SFT remains limited by its offline nature and cannot adapt to novel contexts. Reinforcement learning (RL) has thus emerged as a natural complement, enabling agents to refine policies through trial-and-error interaction and reward-driven adaptation (Zhu et al., 2025; Zhuang et al., 2024). In practice, SFT often serves as initialization, with RL providing fine-grained policy improvement under feedback (Zhu et al., 2025; Li, 2019; Zhang et al., 2021).

Recent research highlights three major directions. First, compiling language into structured controllers—such as graphs, code, or plans—allows RL to optimize execution policies over symbolic abstractions rather than raw text (Zhuang et al., 2024; Jia et al., 2025; Zhu et al., 2025). Second, online collaboration is adapted through RL-based task decomposition, communication routing, and role assignment, which allow dynamic coordination beyond static protocols (Zhou et al., 2025; Wang et al., 2024; Xu et al., 2025; Li et al., 2024a). Third, several studies explore learning reasoning policies directly in language space using GRPO or PPO-style updates, often integrating tools or human input when beneficial (Wan et al., 2025; Park et al., 2025; Han et al., 2025; Peiyuan et al., 2024; Feng

et al., 2024). These approaches underscore the importance of credit assignment and reward shaping for aligning emergent behaviors in high-dimensional language action spaces (Wei et al., 2025; Jiang et al., 2025; Alsadat & Xu, 2024; Lin et al., 2025).

Our work is closely aligned with this trajectory but emphasizes a key gap: most existing RL approaches focus on optimizing intra-agent or inter-agent coordination while leaving the system's knowledge boundary fixed. In contrast, we introduce a dual-loop perspective where RL is responsible for learning a metacognitive deferral policy, and expert demonstrations triggered by deferral events fuel continual learning. This integration addresses both immediate decision-making and long-term capability growth, providing a principled path toward genuinely adaptive multi-agent systems.

## B  METHODOLOGY

### B.1  PRELIMINARIES: THE METACOGNITIVE MARKOV DECISION PROCESS

We formalize the dynamics of human–agent collaboration as a **Meta-Cognitive Markov Decision Process (Meta-MDP)**, defined by the tuple $\mathcal{M} = (\mathcal{S}, \mathcal{A}, P, R, \gamma)$. A Meta-MDP provides a principled framework for sequential decision-making where actions correspond to high-level cognitive strategies. Formally, a Meta-MDP is defined by the tuple $(\mathcal{S}, \mathcal{A}, P, R, \gamma)$. At each round $t$ of the multi-agent collaboration, the process unfolds as follows: The state $s_t \in \mathcal{S}$ is a **structured cognitive state representation**, which encapsulates not only the external problem context but also the agent's internal assessment of its own and its peers' current understanding, as we will detail in Section 3.2. Based on this rich state, the agent selects a metacognitive action $a_t$ from a discrete action space $\mathcal{A}$, which includes functional strategies such as solving the problem autonomously or deferring to the expert. The system then transitions to a new state $s_{t+1}$ according to the transition function $P(s_{t+1}|s_t, a_t)$. A reward $R(s_t, a_t)$ is issued, designed to incentivize both task success and the efficient utilization of the expert resource. The overarching goal is to learn an optimal metacognitive policy $\pi^*(a_t|s_t)$ that maximizes the expected cumulative reward, thereby training an agent that can rationally balance autonomous problem-solving with strategic reliance on human guidance.

## C  EXPERIMENT

### C.1  EXPERIMENTAL SETTINGS

**Benchmarks and Evaluation.**   To evaluate our framework, we conduct experiments on a broad collection of benchmarks that test complementary aspects of reasoning ability. These tasks span three domains: general knowledge and analytical reasoning, program synthesis, and mathematical problem solving.

For general knowledge, we use the MMLU benchmark, which includes 57 subject areas in a multiple-choice format. Performance is measured by Accuracy. To control evaluation cost, we sample 10 examples from each subject category. For program synthesis, we adopt HumanEval, where models generate code solutions from natural-language specifications.

For quantitative reasoning, we consider four math-focused datasets with concise numerical answers: GSM8K (grade-school arithmetic word problems), AIME (short-form olympiad-style tasks), and AMC (contest problems). Performance on these datasets is reported as Solve Rate, defined by exact match against each dataset's normalized reference solution.

All evaluations are conducted on the official datasets with standard prompting protocols. We exclude external tools and retrieval, ensuring that improvements stem from our collaboration framework rather than auxiliary resources. In single-agent settings, inference is run deterministically. For multi-agent experiments involving stochastic sampling, we set up random seeds, repeat runs, and report averaged results. This setup isolates the contribution of our proposed method and allows for fair comparison against existing approaches.

**Baselines.**   To ensure a fair and comparison comparison, we evaluate our framework against three broad families of baseline methods that represent the dominant paradigms in collaborative reasoning with LLMs:

1. **Single-Agent Solvers.** These methods rely on a single model instance without peer interaction. They capture the performance limits of prompting alone. Examples include direct decoding under a standard prompt (*Vanilla*), reasoning traces generated by *Chain-of-Thought (CoT)* prompting, and multi-sample aggregation methods such as *Self-Consistency (SC)*. Self-reflection strategies (e.g., *Reflection*, *RASC*) are also included, where the model internally revises its outputs without external assistance.

2. **Interactive Multi-Agent Deliberation.** This class introduces explicit communication among multiple agents. Agents generate, critique, and refine one another's proposals. Approaches such as *LLM-Debate* implement structured argue–respond cycles, while pairwise or pooled critique frameworks (e.g., *PHP*) simulate peer-review processes. These baselines assess whether systematic interaction alone, without external expertise, can reduce errors and improve reasoning robustness.

3. **System-Level Coordination Frameworks.** Some approaches treat collaboration as an optimization problem over computational graphs. Adaptive topology and routing methods (e.g., *DyLAN*, *MasRouter*) dynamically determine communication patterns, while workflow- and search-based systems (e.g., *GPTSwarm*, *AFLOW*) orchestrate reusable reasoning modules. Communication-pruning strategies such as *AgentPrune* improve scalability by filtering redundant interactions. These baselines highlight efficiency and coordination at scale.

For all baselines, we control the backbone model, prompting setup, and generation budget (number of agents, rounds, and outputs). When multiple candidates are produced, we apply the baseline's canonical reduction method (e.g., majority vote). No retrieval augmentation or external tools are used. This categorization clarifies whether improvements arise from stronger *single-agent reasoning*, richer *peer verification*, or more effective *coordination*, providing a clear context for evaluating our method.

**Implementation Details.** Unless otherwise specified, our main experiments are conducted with a three-agent collaborative setup running for three interaction rounds. All autonomous agents are instantiated from the instruction-tuned open-source backbones **Qwen2.5-7B-Instruct**, **Qwen2.5-3B-Instruct** (Team, 2024), **Llama-3.1-8B-Instruct**, and **Llama-3.2-3B-Instruct** (Grattafiori et al., 2024). To enable efficient post-training while controlling memory overhead, we adopt parameter-efficient fine-tuning via LoRA, with a default rank of 16, scaling factor $\alpha = 32$, and dropout rate 0.05. In practice, we implement the system using the HuggingFace Transformers framework, together with 8-bit quantization and key–value caching to reduce memory consumption and accelerate multi-round inference. For autonomous agent generation, decoding follows a nucleus sampling strategy with top-$p = 0.95$, temperature 0.7, and a maximum generation length of 1024 tokens. For the external GPT-based human proxy, we use a lower temperature of 0.3 to encourage more stable and reliable responses during intervention. To reduce randomness in evaluation, most main experiments are repeated with three independent random seeds and the reported results are averaged across runs.

For optimization, we use Adam with cosine learning-rate decay. In our hyperparameter search, the learning rate is selected from $\{2 \times 10^{-5}, 5 \times 10^{-5}, 1 \times 10^{-4}, 2 \times 10^{-4}, 5 \times 10^{-4}\}$, and the number of training epochs is varied within the range 1–3. We further control the effective batch size through gradient accumulation, with batch-size settings explored in $\{8, 16, 32\}$. For the GRPO-style inner-loop optimization, we use a clipping coefficient $\epsilon = 0.2$, a KL regularization weight $\beta_{\text{kl}} = 0.02$, reward scaling fixed to 1.0, and advantage normalization enabled by default. These settings are chosen to maintain stable offline policy optimization while preserving sufficient sensitivity to relative reward differences across candidate actions.

The training data consist of both real and synthetic task instances. For GSM8K, we directly use the original training split. For the remaining benchmarks, in order to better simulate realistic help-seeking scenarios in which agents may need to request external assistance, we construct synthetic training data with GPT-5, generating task-aligned samples that mimic the types of situations encountered during collaborative reasoning. In total, we use approximately 10K training instances across all tasks. Data generation and offline trajectory construction are performed under the HILA framework. Specifically, for each agent state, we enumerate all valid strategic actions and roll out the corresponding consequences to form grouped candidates, which are then converted into contrastive comparisons for offline GRPO training. This exhaustive enumeration allows us to build

action groups with explicit relative preferences, rather than relying only on the single action sampled during online execution.

In addition to the policy-training data, we also construct supervised data for the outer-loop continual learning stage. Whenever an agent chooses to seek external help, we record the corresponding problem together with the expert-provided correct reasoning trace and final answer, and treat this pair as a ground-truth supervision instance for SFT. After preprocessing, the resulting offline dataset contains roughly 8K grouped samples for GRPO training and about 2K expert-labeled samples for SFT. This split reflects the intended role of the dual-loop design: the inner loop is trained primarily from action-level relative comparisons, while the outer loop is driven by a smaller but higher-quality set of expert demonstrations that directly expand the model's reasoning capability.

For reward shaping, we fix the overall reward scale to $1.0$ and tune the penalties associated with different strategic actions. In particular, the penalty for CREATE is searched over $\{0.05, 0.1, 0.2\}$, while the penalty for DEFER is varied from $0.1$ to $0.5$ with step size $0.1$. These coefficients control the trade-off between autonomous exploration, reliance on peer solutions, and external intervention. We tune them on held-out validation performance to balance reasoning accuracy against the cost of invoking additional computation or expert assistance. Across all experiments, we use the same implementation pipeline and tuning protocol unless a specific ablation explicitly studies one of these hyperparameters.

**Definition of Human Expert.** In principle, the human collaborator in our framework refers to a real person who can provide external knowledge and corrective interventions. However, following prior studies that approximate human input with advanced language models (Liu et al., 2023; Pandya et al., 2024), we also adopt strong LLMs as practical substitutes. For **LLaMA**-based agents, whose reasoning ability is relatively weaker, we use **GPT-4o-mini** as the proxy expert in most experiments to maintain a favorable balance between cost and effectiveness. In contrast, for **Qwen**-based agents, whose reasoning ability is substantially stronger, a more powerful expert is necessary to provide sufficiently informative guidance; therefore, we use **GPT-4o** as the proxy expert in those settings.

**Prompt Templates.** Our framework relies on a small set of task-specific and interaction-specific prompt templates to coordinate reasoning, collaboration, and metacognitive control. Concretely, we use base prompts to initialize task solving under different task types (e.g., mathematical reasoning, multiple-choice question answering, and code generation), a debate prompt to support iterative multi-agent refinement by exposing each agent to its own prior response and the responses of other agents, and a meta-policy prompt to govern high-level action selection among EVAL, CREATE, and DEFER. For clarity, we present the canonical forms of these prompts below, with placeholder variables indicating the task content and the dynamically constructed interaction history.

---

**Base Prompt (Multiple-Choice)**

Answer the following multiple-choice question. Choose the single best option.
Solve the problem:
`{{question}}`
Think step by step, show your reasoning, and end your response with a single line that clearly states the final answer.

---

**Base Prompt (Math: Numeric / Symbolic)**

Solve the following problem:
`{{question}}`
Think step by step, show your reasoning, and be careful with arithmetic. End your response with a single line that clearly states the final answer. If the answer is a number, output only the number on that final line.

---

---

**Base Prompt (Math with Forced Boxed Answer)**

Solve the following problem:
`{{question}}`
Think step by step, show your reasoning, and be careful with arithmetic. Must give the final answer in the form `\boxed{...}`.

---

**Base Prompt (Code Generation with Unit Tests)**

You are given a Python programming task. Write a correct and efficient solution that passes all unit tests.
Rules:

- Output ONLY Python code.
- Do NOT include explanations, comments outside the given prompt, or additional text.
- Keep the original function signature exactly as given.
- Do not write any test code. Do not use `input()` or `print()`.
- You may use the Python standard library.

`{{question}}`

---

**Base Prompt (Generic Fallback)**

Solve the following problem:
`{{question}}`
Think step by step. End your response with a single line that clearly states the final answer. If the answer is a number, output only the number on that final line.

---

**Multi-agent Collaboration Prompt**

You are in a multi-agent collaboration.
**=== Original Prompt ===**
`{{base_prompt}}`
**=== Your Previous Responses ===**
`{{self_history_block}}`
**=== Other Agents' Responses ===**
`{{others_history_block}}`
Now compare the solutions, resolve disagreements, and provide an UPDATED final answer. Keep reasoning concise but correct. Finish with a clear final answer line.

---

**Meta-Policy Prompt**

You are a meta-policy controller for a multi-agent system. Choose ONE action and output ONLY the action line.
**Valid actions (no extra text):**

- `DEFER`                                              (ask a human expert)
- `EVAL <idx>`                          (copy Agent idx; idx in `0..N-1`)
- `CREATE`                                  (write a new solution yourself)

`{{structured_decision_signals}}`
**=== Problem ===**
`{{base_prompt}}`
**=== Your Latest Solution ===**
`{{self_latest_solution}}`

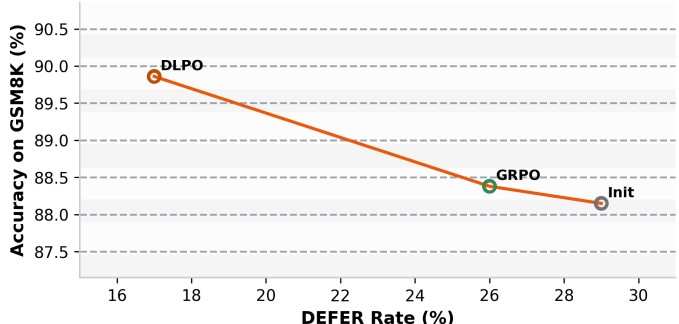

Figure 4: Accuracy vs. DEFER rate on GSM8K across training stages (Init → GRPO → DLPO). DLPO achieves a joint improvement, increasing accuracy while reducing reliance on external intervention.

---

**=== Other Agents' Latest Solutions ===**
`{{others_latest_solutions}}`
Now output ONLY one action line.

---

## C.2 EXPERIMENTAL RESULTS

**Accuracy–Deferral Coupling Across Training Stages.** To further probe whether the reduced DEFER rate reflects genuine capability gains (rather than merely cost-avoidance), we jointly analyze task accuracy and deferral frequency across training stages. Figures 4–6 reveal a consistent and interpretable pattern: moving from the initialization policy to GRPO and then to full DLPO generally shifts the system toward the upper-left region, i.e., *higher accuracy with lower reliance on external intervention*. This coupling is especially informative because it distinguishes two qualitatively different regimes: (i) a *frugal* policy that reduces deferral but sacrifices correctness, and (ii) a *stronger* policy that reduces deferral *while* improving accuracy. Across benchmarks, GRPO tends to yield a modest leftward shift (lower DEFER), consistent with learning a cost-aware intervention policy. In contrast, DLPO produces a clearer Pareto-style improvement on all three datasets, simultaneously decreasing deferral and increasing accuracy, which is difficult to explain by cost-awareness alone. Instead, it is consistent with the dual-loop design: the inner loop improves *when* to ask, while the outer loop assimilates expert demonstrations so that the system increasingly *does not need to ask*. Notably, the magnitude of this coupled improvement varies by task: on GSM8K, the accuracy gains are relatively modest while deferral decreases substantially, suggesting that DLPO primarily reduces unnecessary escalation on easier arithmetic items; on AMC, GRPO alone exhibits a mild trade-off, whereas DLPO recovers and improves accuracy while further reducing deferral, indicating that continual learning helps avoid "over-pruning" intervention; and on MMLU, the monotonic improvement of both metrics provides strong evidence that the learned policy becomes both more selective and more competent. Overall, these coupled trajectories provide an additional behavioral validation that HILA's gains arise from both calibrated intervention and capability growth.

**Effect of Human Proxy Capability.** We further examine how the capability of the external expert affects the overall performance of HILA by replacing the human proxy with language models of different strengths. As shown in Figure. 2 and Table. 6, The results show a clear and consistent pattern: stronger human proxies lead to better final performance across all evaluated benchmarks. When the proxy is weaker, the gains from intervention are noticeably reduced, whereas a stronger proxy produces the best overall results. This trend is consistent with the design of HILA, where DEFER is intended to provide access to higher-quality reasoning trajectories whenever the autonomous collective reaches its limits.

These results suggest that the effectiveness of HILA depends not only on learning *when* to invoke external help, but also on the quality of the guidance received once intervention occurs. In other words, strategic deferral and expert quality are complementary: a well-trained metacognitive policy

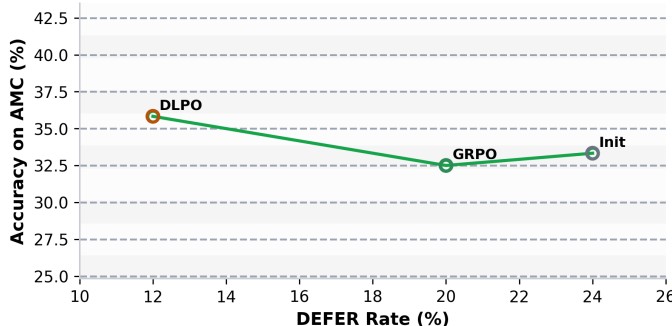

Figure 5: Accuracy vs. DEFER rate on AMC across training stages. GRPO reduces deferral with a small accuracy fluctuation, while DLPO simultaneously improves accuracy and further lowers deferral, suggesting that continual learning mitigates over-conservative intervention.

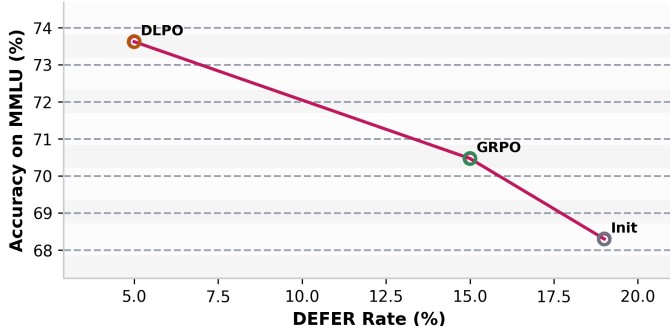

Figure 6: Accuracy vs. DEFER rate on MMLU across training stages. Accuracy increases monotonically as deferral decreases, indicating that the learned policy becomes both more selective and more capable.

can identify situations where outside assistance is valuable, but the ultimate benefit still depends on how informative and reliable that assistance is. Overall, this experiment shows that HILA can effectively benefit from stronger external guidance when available, while also highlighting the importance of the chosen human proxy in determining final system performance.

**Deferral Cost as a Control Knob for Collaboration.** We further examine how the deferral cost $C_{defer}$ modulates the balance between autonomous reasoning and external intervention in HILA. Figure 7(a) shows the accuracy trends on AMC and MMLU under different values of $C_{defer}$, while Figure 7(b) illustrates how the action distribution changes on AMC when comparing a low cost 0.1 with a high cost 0.5. A clear pattern is that increasing $C_{defer}$ pushes the policy away from DEFER and toward autonomous actions, especially EVAL. This confirms that the learned policy responds directly to the intended intervention penalty and adjusts its behavior accordingly. At the same time, the accuracy curves reveal a nontrivial trade-off. A moderate cost can reduce unnecessary reliance on external help while preserving competitive performance, whereas a larger cost suppresses deferral more aggressively but may also hurt final accuracy. This effect is particularly visible on the more difficult benchmark, where overly discouraging intervention can prevent the system from accessing useful external guidance when it is still needed. These results support the view that $C$ serves as an effective control knob for collaboration, regulating how often HILA relies on outside expertise while exposing the performance–cost trade-off that must be tuned in practice.

**The Value of Collective Exploration.** To examine how the size of the collective influences both effectiveness and cost, we evaluate HILA with varying numbers of autonomous agents on AMC and MMLU, as shown in Table. 7 and Table. 8. The results show a broadly consistent trend across both benchmarks: increasing the number of agents generally improves accuracy, especially when moving

| Model | GSM8K | AMC | MMLU |
|---|---|---|---|
| HILA with gpt-4o-mini | 88.15 | 33.33 | 68.30 |
| HILA with gpt-3.5-turbo | 83.33 | 17.50 | 66.84 |
| HILA with gpt-4o | **89.76** | **36.67** | **72.28** |

Table 6: Effect of human proxy capability on HILA. Using stronger language models as the external expert consistently improves performance on GSM8K, AMC, and MMLU, showing that the quality of external guidance is an important factor in the effectiveness of strategic deferral.

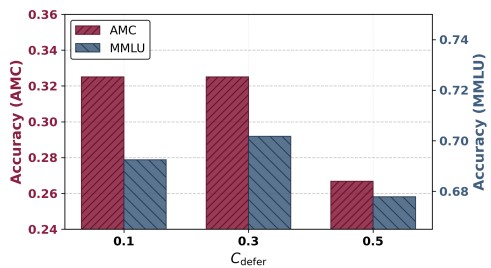

(a) Accuracy on AMC and MMLU under different deferral costs $C_{defer}$.

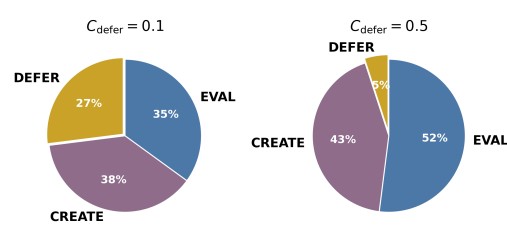

(b) Action distribution on AMC for low and high deferral costs (0.1 vs. 0.5).

Figure 7: Deferral cost as a control knob for collaboration. (a) shows how changing the deferral cost $C$ affects task accuracy on AMC and MMLU. (b) shows how increasing cost shifts the learned policy on AMC away from DEFER and toward autonomous actions, especially EVAL. Together, the results illustrate the trade-off between reducing external intervention and preserving final performance.

from a very small collective to a moderate-sized one. This pattern supports a central premise of our framework, namely that larger agent pools enable richer *collective exploration*. With more agents generating and refining candidate solutions in parallel, the system can access a broader set of reasoning trajectories, increasing the chance that at least one high-quality solution emerges and can be selected or propagated through the metacognitive policy.

At the same time, the gains are not linear. Accuracy improves most clearly in the low-agent regime, but becomes less stable as the collective grows larger, with smaller incremental gains and occasional fluctuations across neighboring settings. This suggests diminishing returns from simply scaling the number of agents: once the solution space has been sufficiently diversified, additional agents contribute progressively less new information. In contrast, computational cost increases sharply with the agent count. Both input and output tokens grow rapidly as more agents participate, leading to a substantial increase in total token usage even when accuracy improves only marginally. Taken together, these results highlight a practical trade-off: larger collectives can strengthen reasoning through broader exploration, but the benefit must be balanced against the rising inference cost. This makes the number of agents an important efficiency–performance design choice in deploying HILA.

**Optimal Depth of Iterative Collaboration.** To assess the role of iterative refinement, we vary the number of collaborative rounds and evaluate HILA on AMC and MMLU, as shown in Table 9 and Table 10. The results show a clear non-monotonic pattern across both benchmarks. Increasing the number of rounds from a shallow interaction depth to a moderate one consistently improves accuracy, indicating that multi-round collaboration helps the system refine intermediate solutions, incorporate peer feedback, and recover from errors made in earlier stages. This behavior supports the intuition that iterative interaction can strengthen reasoning by allowing agents to progressively revise and consolidate candidate solutions rather than relying on a single-pass exchange.

However, the benefit of additional rounds does not continue indefinitely. Performance peaks at an intermediate depth and then declines slightly when the interaction is extended further, even as token usage continues to increase. This suggests that while additional rounds initially provide useful opportunities for correction and refinement, excessive interaction can introduce diminishing returns and may even begin to reinforce suboptimal trajectories. One plausible explanation is that repeated

| # Agents | AMC | Avg. Input Tokens | Avg. Output Tokens | Avg. Total Tokens |
|---|---|---|---|---|
| 1 | 26.67 | 3146 | 1299 | 4445 |
| 2 | 28.33 | 10184 | 2564 | 12748 |
| 3 | 32.50 | 23285 | 4463 | 27748 |
| 4 | 31.67 | 42540 | 6349 | 48889 |
| 5 | 33.33 | 64097 | 7585 | 71682 |
| 6 | 32.50 | 92135 | 9501 | 101636 |
| 8 | 34.17 | 162142 | 12704 | 174846 |
| 10 | 35.83 | 238278 | 14928 | 253206 |

Table 7: Effect of the number of agents on performance and token cost on AMC. Increasing the collective size generally improves accuracy, but also leads to a steep rise in input, output, and total token usage, revealing a clear performance–efficiency trade-off.

| # Agents | MMLU | Avg. Input Tokens | Avg. Output Tokens | Avg. Total Tokens |
|---|---|---|---|---|
| 1 | 66.32 | 2054 | 508 | 2562 |
| 2 | 67.54 | 5575 | 972 | 6547 |
| 3 | 68.49 | 11049 | 1525 | 12574 |
| 4 | 68.95 | 18302 | 2077 | 20379 |
| 5 | 68.60 | 27133 | 2609 | 29742 |
| 6 | 69.47 | 38317 | 3223 | 41540 |
| 8 | 69.12 | 65295 | 4333 | 69628 |
| 10 | 69.65 | 94827 | 5612 | 100439 |

Table 8: Effect of the number of agents on performance and token cost on MMLU. A larger collective typically yields better accuracy, especially in the low-agent regime, while substantially increasing token consumption as the system scales.

exchanges increase the chance of propagating noisy or flawed intermediate reasoning, making the collective more vulnerable to error accumulation rather than correction. Taken together, these results indicate that iterative collaboration is valuable but not unbounded: the number of rounds should be chosen to balance the gains from refinement against the rising computational cost and the risk of over-iteration.

### C.2.1 REAL HUMAN-IN-THE-LOOP

**Motivation and Evaluation Scope.** Our main experiments use strong GPT-based models as proxies for external experts, which provides a controlled and scalable approximation of human intervention. However, the central claim of HILA is fundamentally about *human–agent* collaboration rather than model–model interaction alone. To validate that the framework remains effective when the external expert is replaced by real human participants, we conduct an additional human-in-the-loop study with real PhD-level experts. Because real human annotations are substantially more expensive than API-based proxy supervision, we focus on three representative benchmarks that jointly cover arithmetic reasoning, competition-style mathematical problem solving, and general knowledge reasoning: GSM8K, AMC, and MMLU. This selection provides diversity in task format and difficulty while keeping the study practically feasible. Accordingly, we instantiate real human involvement in two operational forms: *proactive* guidance provided at initialization and *reactive* assistance, as summarized in Figure 8.

**Balanced Subset Construction for Human Evaluation.** To ensure that the human-evaluation subset faithfully reflects the range of collaborative states encountered by HILA, we do not simply sample instances uniformly at random. Instead, for GSM8K and MMLU, we construct a stratified subset based on the *pre-intervention agreement pattern* among agents, so that the selected instances cover qualitatively different collective reasoning regimes. Concretely, each item is assigned to one of the following categories: (i) *all agents agree and are correct*, (ii) *the majority agrees but is incorrect*, (iii) *the majority agrees and is correct*, and (iv) *all agents disagree*. We then approximately balance

| # Rounds | AMC | Avg. Input Tokens | Avg. Output Tokens | Avg. Total Tokens |
|---|---|---|---|---|
| 1 | 18.33 | 455 | 2195 | 2650 |
| 2 | 22.50 | 13406 | 3480 | 16886 |
| 3 | 31.67 | 24129 | 4580 | 28709 |
| 4 | 40.83 | 33287 | 5203 | 38490 |
| 5 | 35.00 | 43984 | 6627 | 50611 |

Table 9: Effect of the number of collaborative rounds on performance and token cost on AMC. Accuracy improves substantially as the interaction depth increases from one to four rounds, but declines when an additional round is added, indicating diminishing returns from over-iteration.

| # Rounds | MMLU | Avg. Input Tokens | Avg. Output Tokens | Avg. Total Tokens |
|---|---|---|---|---|
| 1 | 0.6404 | 5352 | 933 | 6285 |
| 2 | 0.6491 | 6385 | 1251 | 7636 |
| 3 | 0.6849 | 11049 | 1525 | 12574 |
| 4 | 0.6930 | 15318 | 1798 | 17116 |
| 5 | 0.6895 | 19689 | 2144 | 21833 |

Table 10: Effect of the number of collaborative rounds on performance and token cost on MMLU. A moderate number of rounds improves accuracy, while further increasing the interaction depth yields only limited benefit and slightly higher cost.

these categories in the final evaluation subset. This stratification is important because it exposes human experts to both easy and hard collaborative contexts, including cases of consensus failure and severe disagreement, rather than over-representing only trivial instances. For AMC, since the benchmark itself is comparatively small, we evaluate on the full set instead of a stratified subset.

**Human Expert Pool and Participation Protocol.** As shown in Figure 9, we recruit 20 PhD students from a broad range of technical disciplines, including Computer Science, Computer Engineering, Electrical Engineering, Mathematics, Industrial and Systems Engineering, Biomedical Engineering, Earth Sciences, and Economics, and treat them as domain-general expert annotators. To make real human participation compatible with repeated and reproducible evaluation, we adopt an *offline expert collection* protocol: each human expert answers the selected questions in advance, rather than being queried interactively during live inference. For every problem, each expert provides two fields: `human_idea`, which contains a concise problem-solving intuition or high-level plan, and `human_reasoning`, which contains a more complete reasoning trace together with the final answer. This design allows us to evaluate both lightweight human hints and full expert solutions under the same system. Formally, if $x$ denotes the input problem and $h$ denotes a human expert, then the collected annotation can be written as

$$\mathcal{E}_h(x) = \left( e_h^{\text{idea}}(x),\ e_h^{\text{reason}}(x) \right),$$

where $e_h^{\text{idea}}(x)$ and $e_h^{\text{reason}}(x)$ denote the two levels of expert guidance.

**Two Modes of Human Participation.** We study two distinct modes of real human involvement under HILA. In the first mode, the human serves as a **reactive expert**, i.e., the human is consulted only when an agent selects the DEFER action. In this case, the human response functions exactly as the external expert signal in the standard HILA loop. In the second mode, the human serves as a **proactive expert**, i.e., expert information is injected *before* the agents begin their initial reasoning. This proactive information can be either the concise `human_idea` or the full `human_reasoning`. Importantly, these two roles are not mutually exclusive: even when proactive human information is provided at initialization, the system may still choose to defer later to a GPT-based external expert during downstream collaboration. This lets us separate two complementary effects of human involvement: *initial guidance* (better starting trajectories) and *on-demand correction* (targeted rescue under uncertainty). In notation, the proactive setting augments the original problem input $x$ into

$$x' = \left( x, \mathcal{E}_h^{\text{init}}(x) \right),$$

where $\mathcal{E}_h^{\text{init}}(x)$ may be either $e_h^{\text{idea}}(x)$ or $e_h^{\text{reason}}(x)$.

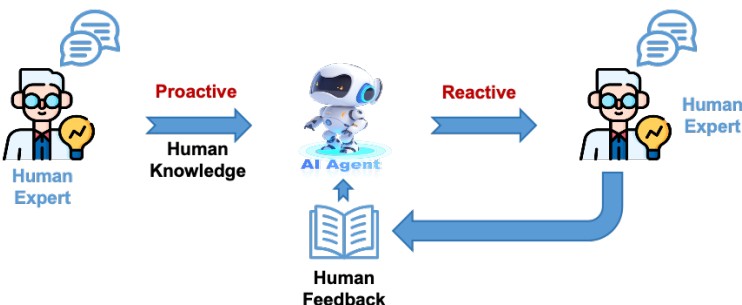

Figure 8: Real human-in-the-loop collaboration in HILA with *bidirectional* interaction. Real human experts can act **proactively** by injecting prior knowledge at initialization (e.g., a high-level `human_idea` or a full `human_reasoning`), shaping the agents' starting state before autonomous collaboration begins. They can also act **reactively** by providing corrective feedback when the meta-policy selects `DEFER`. This unified view highlights that human expertise in HILA is not limited to post-hoc supervision, but can be integrated both as upfront guidance and as on-demand intervention during multi-round collaboration.

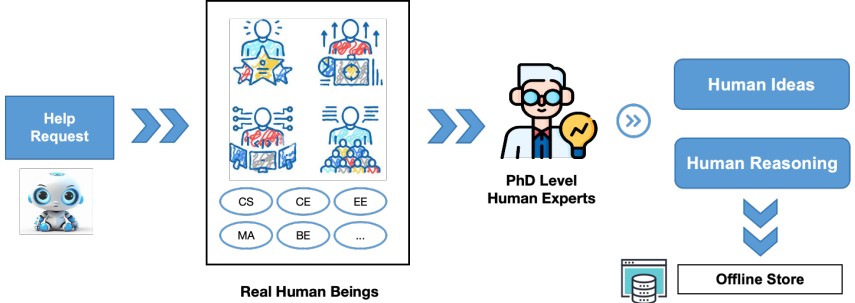

Figure 9: Real human feedback pipeline in HILA. Upon a help request (`DEFER`), the system routes the query to a pool of human experts, who provide either high-level *human ideas* or full *human reasoning*. The collected feedback is then stored as structured supervision for subsequent training and continual improvement.

**Expert Quality in the Single-Expert Regime.** Before evaluating the full HILA framework, we first compare the standalone problem-solving accuracy of candidate experts, including the base LLAMA3-8B model, several GPT models, and the real human expert pool. This serves as a direct estimate of the *quality ceiling* available to the system once external intervention occurs. As shown in Table 11, real human experts achieve the strongest overall performance across the three benchmarks, with especially large gains on AMC, where they substantially outperform all model-based experts. This result is important because it shows that the human expert channel is not merely a conceptual replacement for GPT proxies, but can provide genuinely stronger guidance in domains where structured reasoning and careful mathematical judgment are especially important. At the same time, the margin on MMLU is smaller, suggesting that the relative advantage of human experts depends on task type. Overall, this comparison establishes that real humans constitute a high-quality and meaningful source of external expertise for HILA.

**Reactive Human Experts Under HILA.** We next replace the GPT-based expert in HILA with different reactive experts while keeping the autonomous backbone fixed to LLAMA3-8B. The results in Table 12 show that stronger experts consistently lead to better downstream performance, and real human experts provide the best overall results. This trend mirrors our earlier proxy-based observations, but now with actual human participants: once the system learns *when* to defer, the final gain still depends critically on the quality of the external assistance. The improvement is especially striking on AMC, where replacing GPT experts with real humans yields a much larger jump than on the other two datasets. This suggests that the value of human intervention is particularly strong in settings where the collective may otherwise converge to a plausible but incorrect line of reason-

| Expert | GSM8K | AMC | MMLU |
|--------|-------|-----|------|
| LLaMA3-8B | 58.67 | 16.67 | 57.33 |
| GPT-3.5-Turbo | 66.00 | 17.50 | 58.00 |
| GPT-4o-mini | 85.67 | 39.17 | 86.67 |
| GPT-4o | 88.33 | 46.67 | 87.33 |
| Human Experts | **95.00** | **97.50** | **88.00** |

Table 11: Standalone performance of candidate external experts on the human-evaluation benchmarks. Human Experts denote the aggregated real-expert pool collected from 20 PhD participants. All values are reported as percentages.

| Reactive Expert in HILA | GSM8K | AMC | MMLU |
|-------------------------|-------|-----|------|
| GPT-3.5-Turbo | 61.67 | 20.83 | 59.67 |
| GPT-4o-mini | 74.33 | 29.17 | 64.33 |
| GPT-4o | 77.33 | 33.33 | 67.33 |
| Human Experts | **78.67** | **61.67** | **75.33** |

Table 12: Performance of HILA with different *reactive* experts, i.e., experts consulted only when the system selects `DEFER`. The autonomous backbone is fixed to LLaMA3-8B. All values are percentages.

ing. In other words, human experts are not only more accurate in isolation; when used as reactive intervention targets, they also provide more effective corrective signals for HILA's decision process.

**Proactive Human Guidance at Initialization.** Finally, we examine whether humans can contribute not only as reactive experts but also as *proactive sources of guidance* before the multi-agent reasoning process begins. Table 13 compares several human-participation settings under the same LLaMA3-8B backbone. When humans provide only a high-level idea at initialization, the benefit is mixed: the system shows some improvement on MMLU, but the gains are limited or inconsistent on the more mathematical datasets. This suggests that a coarse human hint may improve the initial search direction, but is often insufficient to reliably steer the full collaborative process toward the correct solution. In contrast, when humans provide a full reasoning trace at initialization, performance improves substantially across all three benchmarks, yielding the strongest results among all compared human-participation settings.

**Reactive vs. Proactive Human Roles.** The contrast between `human_idea` and full `human_reasoning` reveals a key mechanism of real human value in HILA. Lightweight proactive hints primarily act as *weak priors*: they may bias the initial reasoning trajectory, but the downstream agents must still reconstruct the full solution path on their own. Full proactive reasoning, by contrast, provides a much stronger structured signal that can reshape the collaborative search process from the outset. Notably, this proactive human information can still be combined with subsequent GPT-based reactive intervention, creating a hybrid setting in which human knowledge improves initialization while automated expert proxies remain available for later rescue. Taken together, these results suggest that real humans can improve HILA in two distinct but complementary ways: as *reactive experts* that correct failures when the system explicitly requests help, and as *proactive experts* that improve the quality of the initial reasoning state before collaboration unfolds. This provides stronger empirical evidence that HILA is not merely compatible with human involvement in principle, but can meaningfully leverage real human expertise under multiple operational modes.

**Action Allocation Under Real Human Participation.** To better understand how real human involvement changes the behavior of HILA beyond final task accuracy, we further analyze the distribution of strategic actions under different human-participation modes. Specifically, we report the proportion of `EVAL`, `CREATE`, and `DEFER` decisions across GSM8K, AMC, and MMLU when humans act as reactive experts, proactive idea providers, or proactive full-reasoning providers. This analysis is important because the same performance gain can arise from qualitatively different collaboration regimes: one system may improve by becoming more self-reliant, while another may improve by more aggressively invoking external help. By comparing action distributions, we can

| Human Participation Setting | GSM8K | AMC | MMLU |
|---|---|---|---|
| GPT-4o as reactive expert only | 77.33 | 33.33 | 67.33 |
| Human Experts as reactive expert only | 78.67 | 61.67 | 75.33 |
| Human proactive idea + GPT-4o reactive expert | 78.33 | 34.17 | 79.33 |
| Human proactive full reasoning + GPT-4o reactive expert | **90.67** | **65.83** | **86.67** |

Table 13: Comparison of different real human participation modes under HILA. In the proactive setting, human information is injected before the initial agent reasoning, while GPT-4o remains available as the reactive expert for later `DEFER` decisions. All values are percentages.

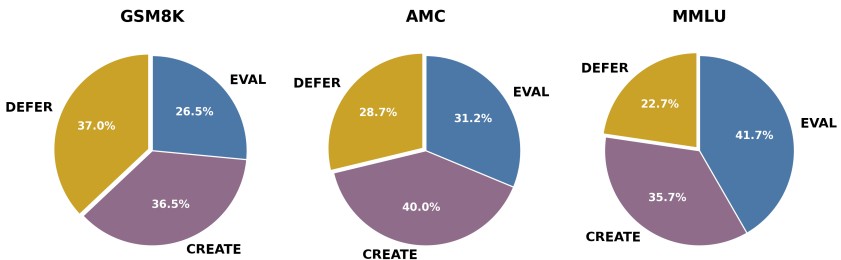

Figure 10: Action distribution under HILA when real humans act as *reactive experts*, meaning that feedback is provided only after the system chooses `DEFER`. Across datasets, `CREATE` and `DEFER` dominate on GSM8K and AMC, while MMLU shows a noticeably larger proportion of `EVAL`.

characterize how the source and form of human input reshape the policy's internal decision dynamics.

**Reactive Human Experts Encourage Direct Help-Seeking.** When real humans serve as *reactive experts*, i.e., they are only consulted after the policy selects `DEFER`, the resulting action distribution shows a substantial allocation to deferral across all three datasets (Figure 10). This is particularly notable on GSM8K, where `DEFER` becomes the most frequent action. The pattern is consistent with the strong standalone quality of human experts observed earlier: once the policy learns that real humans provide highly reliable corrective signals, it becomes more willing to invoke external intervention in uncertain settings. At the same time, the policy still retains nontrivial mass on `CREATE`, suggesting that HILA does not collapse into pure expert reliance even when high-quality human help is available. Instead, it maintains a mixed strategy in which autonomous exploration and expert consultation coexist.

**Proactive Human Ideas Shift the Policy Toward Greater Deferral.** A different pattern emerges when humans provide only a concise `idea` at initialization, while GPT-4o remains available as the reactive expert for later `DEFER` decisions (Figure 11). In this setting, the proportion of `DEFER` increases further across all three benchmarks relative to the reactive-human-only condition. This suggests that a lightweight human idea, although useful as an initial prior, is often insufficient to fully resolve downstream uncertainty. As a result, the policy appears to rely more heavily on later reactive intervention, using the initial hint to guide the search while still frequently requesting stronger expert assistance when difficult reasoning steps remain unresolved. In this sense, high-level human ideas may improve early orientation, but they do not necessarily reduce the model's dependence on external rescue during the full reasoning trajectory.

**Full Proactive Human Reasoning Rebalances the Collaboration Strategy.** When the proactive human input is upgraded from a high-level idea to a full reasoning trace, the action distribution changes again (Figure 12). Compared with the proactive-idea condition, `DEFER` decreases on GSM8K and AMC, while `CREATE` generally becomes more prominent. This pattern suggests that richer upfront human guidance can better stabilize the initial reasoning state, allowing the agents to continue refining or extending a stronger partial solution without needing to invoke reactive help as often. In other words, a full human reasoning trace appears to function less like a weak prior and more like a structured scaffold for downstream collaboration. Although `DEFER` remains non-

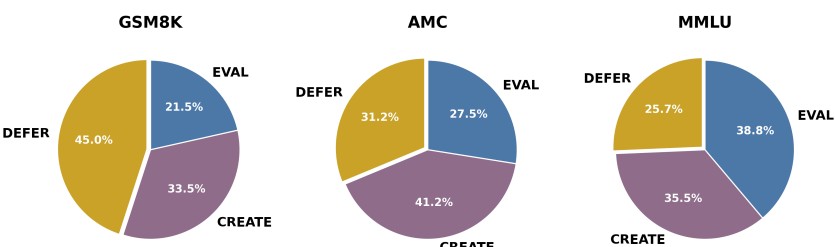

Figure 11: Action distribution under HILA when humans provide a *proactive idea* at initialization, while GPT-4o remains available as the reactive expert for later DEFER decisions. Results are shown for GSM8K, AMC, and MMLU.

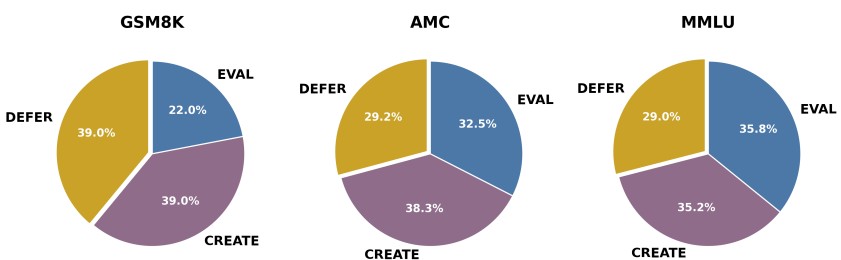

Figure 12: Action distribution under HILA when humans provide *full proactive reasoning* at initialization, while GPT-4o remains available as the reactive expert for later DEFER decisions. Results are shown for GSM8K, AMC, and MMLU.

negligible, the reduction relative to the proactive-idea setting indicates that stronger proactive human input can partially substitute for later intervention by lowering uncertainty earlier in the process.

**Human Guidance Changes Policy Regimes, Not Just Accuracy.** Taken together, these distributions reveal that different forms of human participation induce meaningfully different policy regimes in HILA. Reactive human experts primarily increase the attractiveness of DEFER by making external intervention highly valuable when explicitly requested. Proactive human ideas, while beneficial as initial hints, can still leave enough residual uncertainty that the system defers even more frequently downstream. By contrast, proactive full reasoning provides stronger structural guidance upfront, which reduces the need for subsequent intervention on some tasks and shifts more probability mass toward autonomous continuation through CREATE and EVAL. These observations complement the accuracy results by showing that real human involvement affects not only *how well* the system performs, but also *how* it chooses to collaborate.

## D    DECLARATION ON THE USE OF LARGE LANGUAGE MODELS

In preparing this work, we made use of several large language models for different purposes. First, GPT-4o-mini and GPT-4o were integrated directly into our experimental framework, where they served as proxies for human experts in the human-in-the-loop setting. This design choice follows prior research and allowed us to evaluate the framework under controlled and repeatable conditions while balancing cost and effectiveness. GPT-5 was used to generate synthetic training data for model optimization, enabling scalable and controlled construction of training instances. Second, GPT-5 was employed to assist with improving the clarity, organization, and readability of the manuscript. The model helped refine phrasing and grammar, but all conceptual contributions, methodological design, and experimental analysis were developed by the authors. All content was carefully reviewed, edited, and validated by the authors, who take full responsibility for the accuracy and integrity of the final publication.

