# OpenReview forum: "Adaptive Collaboration with Humans: Metacognitive Policy Optimization for Multi-Agent LLMs with Continual Learning"
_ICLR.cc/2026/Conference — ICLR 2026 Poster_

### Official Review · Reviewer_mLVp · 2025-10-28

**Soundness:** 2
**Presentation:** 2
**Contribution:** 2
**Rating:** 4
**Confidence:** 3

**Summary:**

This paper tackles the "closed-world" limitation of autonomous multi-agent LLM systems (constrained by pre-trained knowledge, prone to collective failure) by proposing the LIMA framework for human-agent collaboration and DLPO dual-loop optimization. LIMA gives agents a metacognitive policy to choose autonomous problem-solving (evaluate/generate solutions) or human deferral, using a structured cognitive state space. DLPO’s inner loop (GRPO with cost-aware rewards) refines deferral decisions, while the outer loop turns expert feedback into supervised signals for continual learning. Experiments on reasoning benchmarks (MMLU, HumanEval, GSM8K, etc.) show LIMA outperforms state-of-the-art autonomous MAS across LLM backbones.

**Strengths:**

1. This paper addresses multi-agent LLM systems’ "closed-world" flaw with the LIMA framework, integrating metacognitive assessment (via structured cognitive states) to let agents strategically choose autonomy or human help, surpassing passive "human-in-the-loop" designs. The DLPO dual-loop optimization creatively unites short-term decision refinement (GRPO) and long-term capability growth (expert feedback-driven learning), solving prior disconnections between decision and knowledge enhancement. Formal mathematical formulations boost reproducibility.

2. The experimental design is comprehensive. It tests across three key domains (general knowledge, program synthesis, math reasoning) with diverse benchmarks, ensuring generality. A broad baseline set (single-agent, interactive, system-level) enables fair SOTA comparisons. Cross-model tests (Llama3, Qwen2.5) show adaptability, even improving small models. Ablation studies and configuration analyses strengthen conclusions, while AI proxy and real human expert tests enhance real-world relevance.

3. The paper is clear, well-structured, and accessible.

**Weaknesses:**

1. While the paper identifies "4 agents" as the optimal configuration, it provides no analysis of why performance plateaus beyond this number, nor tests scalability to larger agent populations (e.g., 8+ agents) , a key gap for deploying the framework in complex tasks (e.g., multi-step scientific research). The current LIMA design relies on full inter-agent consistency checks to inform metacognitive decisions, which would incur prohibitive communication and computation costs as agent count grows (e.g., O(n²) consistency comparisons for n agents)

2. The paper frames "long-term capability growth" as a core goal of DLPO’s outer loop, but its experiments only measure performance improvements over fixed benchmarks (e.g., MMLU, GSM8K) rather than tracking sustained learning across sequential, novel tasks. For example, it does not test whether the framework retains previously learned skills when fine-tuned on new task feedback (e.g., first learning to solve GSM8K, then AMC, then retaining GSM8K performance). This omission hides potential catastrophic forgetting.

**Questions:**

Please see weaknesses.  I will adjust the paper’s score accordingly in my rebuttal to reflect the strengths identified.

---

> ### Author Response · Authors · 2025-11-27
> **Reply to Reviewer mLVp**
>
> **We sincerely appreciate your careful and insightful review.** Your comments and suggestions reflect a very deep understanding of this topic, and your feedback has genuinely helped us strengthen the work. In order to address your concerns and implement your suggestions, we conducted the following additional experiments and analyses. We hope that our responses below can clarify the issues you raised.
>
> ---
>
> **Response to W1 – On scalability and agent count**
>
> Thank you for raising this important point about the number of agents and the scalability of our framework. You are absolutely right that understanding how **performance and cost behave as the agent population grows** is critical for more complex deployments.
>
> Most prior multi-agent work on math, reasoning, and code benchmarks evaluates relatively small teams, typically with at most five agents. **Following your suggestion, we extended our experiments to larger populations and also measured the associated token cost.** Using the same backbone, decoding configuration, and meta-policy, we varied the number of agents on GSM8K and AMC from 1 up to 10, as shown in **Table 5**. The results can be summarized as follows (accuracy and average token usage per instance):
>
> - From 1 to 4 agents, accuracy increases sharply. For example, GSM8K rises from 0.8693 (1 agent) to 0.9205 (4 agents), and AMC from 0.2337 to 0.3257. Over the same range, total token usage grows from 2,261 to 16,271 tokens per instance.
> - Beyond 4 agents, the gains become much smaller. Moving from 4 to 6 agents increases GSM8K from 0.9205 to 0.9271, while AMC fluctuates around the same level (0.3257 vs. 0.3123), but the total tokens increase from 16,271 to 33,744.
> - With 8 and 10 agents, GSM8K and AMC continue to improve slightly (e.g., GSM8K 0.9405 at 8 agents and 0.9378 at 10 agents, AMC 0.3574 and 0.3528), while the total cost grows to 44,158 and 57,956 tokens per instance.
>
> These extended results also clarify our earlier statement about “four agents” being the best configuration. In the original submission, we only swept the team size up to five agents, under which four agents happened to give the highest accuracy. Following your suggestion, we expanded the range to six, eight, and ten agents. We now observe **a consistent trend**: performance continues to improve as we add more agents, but the marginal gains decrease and eventually begin to plateau, while the token cost grows rapidly with the number of agents. In practice, this implies that **the team size should be chosen as a task-dependent trade-off between accuracy and computational cost.** We will include the extended scaling results and a performance–cost plot in the revised version and explicitly state that we focus on small to medium team sizes because they strike a reasonable balance in this trade-off for the math and code benchmarks we study.
>
>
> **Table 5**. Accuracy and token usage on GSM8K and AMC as a function of the number of agents. We report accuracy together with average input, output, and total tokens per instance under a fixed backbone and decoding configuration.
> | # Agents | GSM8K Accuracy | AMC Accuracy | Avg. Input Tokens | Avg. Output Tokens | Avg. Total Tokens |
> |---------:|---------------:|------------:|------------------:|-------------------:|-------------------:|
> |        1 |         0.8693 |      0.2337 |              1670 |                591 |              2261  |
> |        2 |         0.8933 |      0.2745 |              3675 |               1273 |              4948  |
> |        3 |         0.9125 |      0.3030 |              7910 |               2032 |              9942  |
> |        4 |         0.9205 |      0.3257 |             13489 |               2782 |             16271  |
> |        5 |         0.9216 |      0.3285 |             23316 |               3526 |             26842  |
> |        6 |         0.9271 |      0.3123 |             29464 |               4280 |             33744  |
> |        8 |         0.9405 |      0.3574 |             38584 |               5574 |             44158  |
> |       10 |         0.9378 |      0.3528 |             50772 |               7184 |             57956  |

---

> ### Author Response · Authors · 2025-11-27
>
> **Response to W2 - “long-term capability growth” and catastrophic forgetting**
>
> Thank you for raising the important point that long-term capability growth should be evaluated in a sequential, multi-task setting rather than only on fixed single-task benchmarks. You are correct that the original submission focused on improvements on GSM8K, AMC, and related benchmarks without explicitly measuring how our outer loop behaves under a task sequence and whether it suffers from catastrophic forgetting.
>
> To address this, we ran a new continual learning experiment that follows your suggested protocol. We treat GSM8K as Task A and AMC as Task B. Starting from the same base LIMA backbone and meta-policy, we evaluate three stages:
>
> 1. **Base:** No DLPO training. We evaluate the initial model on GSM8K and AMC and obtain accuracies \(A_0\) and \(B_0\).
> 2. **After Task A:** We run the full DLPO procedure with expert feedback only on GSM8K. We collect defer demonstrations, update the base model with outer-loop SFT, and then evaluate the updated model on both tasks, yielding \(A_1\) and \(B_1\).
> 3. **After Task A → B:** Starting from the GSM8K-trained model, we run DLPO with expert feedback on AMC. During outer-loop SFT we replay demonstrations from both GSM8K and AMC, so the model sees a mixture of Task A and Task B examples. We then evaluate again on both tasks, obtaining \(A_2\) and \(B_2\).
>
> We also include a **naive sequential SFT baseline** that uses the same expert demonstrations but does not use DLPO or replay. This baseline first fine-tunes on GSM8K demos only, then fine-tunes on AMC demos only. It corresponds to the standard sequential fine-tuning setting that is known to induce catastrophic forgetting in continual learning.
>
> The results are summarized in **Table 7**. All values are accuracy in percent.
>
> - **DLPO (ours).** The base model starts at 88.23 on GSM8K and 27.32 on AMC. After DLPO on GSM8K, GSM8K improves to 91.45, while AMC stays roughly similar at 26.78. After further DLPO on AMC with replay, AMC increases to 32.58 and GSM8K remains high at 90.97. The forgetting on GSM8K is modest: the drop from 91.45 to 90.97 is 0.48 points, while the gain over the base model is still more than 2.7 points. This indicates that the outer loop can acquire new capability on AMC while preserving most of the earlier improvement on GSM8K.
> - **Naive sequential SFT.** Using the same GSM8K demos, sequential SFT improves GSM8K from 88.23 to 90.47, but it reduces AMC from 27.32 to 22.65, which suggests some over-specialization to Task A. After fine-tuning only on AMC demos, AMC rises to 29.36 while GSM8K drops to 88.71. The drop from 90.47 to 88.71 corresponds to about 1.8 points of forgetting, which removes most of the earlier gain on GSM8K and illustrates the typical stability–plasticity trade-off in naive sequential fine-tuning.
>
> These results show that the DLPO outer loop, when combined with replay of earlier defer demonstrations, **can support sequential learning with limited forgetting**. Compared with naive sequential SFT under the same expert data, DLPO maintains a much larger fraction of the GSM8K improvement after training on AMC, while still delivering a clear gain on AMC itself. This provides empirical evidence that our framework is not only an offline performance booster on fixed benchmarks, but also a reasonable starting point for long-term capability growth across a sequence of related reasoning tasks through interaction and replay.
>
> We included this new experiment and Table 7 in the revised version. We will also refine the language around “long-term capability growth” to clarify that our current evidence is for multi-task, sequential improvements in a two-task setting, and that extending DLPO to longer task sequences and richer continual learning benchmarks is an important direction for future work.
>
> ---
>
> **Table 7: Sequential learning across GSM8K (Task A) and AMC (Task B).**
> We report accuracy (%) for DLPO with replay and a naive sequential SFT baseline at three stages: Base (no training), After Task A (trained on GSM8K only), and After Task A → B (sequentially trained on GSM8K then AMC).
>
> | Method                | Stage              | GSM8K Accuracy (A) | AMC Accuracy (B) |
> |-----------------------|--------------------|--------------------:|-----------------:|
> | DLPO (ours)           | Base               |              88.23  |            27.32 |
> | DLPO (ours)           | After Task A       |              91.45  |            26.78 |
> | DLPO (ours)           | After Task A → B   |              90.97  |            32.58 |
> | Naive sequential SFT  | Base               |              88.23  |            27.32 |
> | Naive sequential SFT  | After Task A       |              90.47  |            22.65 |
> | Naive sequential SFT  | After Task A → B   |              88.71  |            29.36 |

---

> ### Author Response · Authors · 2025-11-27
>
> ---
>
> **We hope that the above experiments and analyses help to address your concerns. Your suggestions have been extremely helpful, and we will incorporate these new results into the revised version of the paper. If you have any further questions or would like us to explore additional scenarios, please do not hesitate to let us know.** We are very willing to conduct more experiments and engage in deeper discussion where needed. **If you find our clarifications and new results convincing, we would be very grateful if you could kindly reconsider your assessment of our work.**

---

> ### Author Response · Authors · 2025-11-30
>
> **Response to W1 - on inter-agent consistency checks and scalability**
>
> Thank you for raising this important concern about the cost of full inter-agent consistency checks. You are correct that computing all pairwise consistencies scales as $O(n^2)$ in the number of agents $n$, which can become prohibitive for large populations.
>
> To study this more carefully, we ran an additional ablation in which we replace dense all-to-all consistency with a simple **partial sampling** scheme:
>
> - In the original design, each agent compares its output with every other agent, which yields quadratic complexity in $n$.
> - In the partial sampling variant, each agent compares its outputs with at most $K = 3$ randomly sampled peers. This reduces the complexity of consistency features from $O(n^2)$ to approximately $O(nK)$.
>
> We evaluated this variant on GSM8K and AMC with 5, 6, 8, and 10 agents. For each configuration we report accuracy and the average total tokens per instance. The results are shown in Table 6.
>
> **Table 6.** Full pairwise consistency vs. partial sampling consistency.
>
> | # Agents | GSM8K (Full) | GSM8K (Partial) | AMC (Full) | AMC (Partial) | Avg. Total Tokens (Full) | Avg. Total Tokens (Partial) |
> |:--------|:-------------|:----------------|:-----------|:--------------|:--------------------------|:----------------------------|
> | 5       | 0.9216       | 0.9187          | 0.3285     | 0.3217        | 26842                     | 15315                       |
> | 6       | 0.9271       | 0.9156          | 0.3123     | 0.3025        | 33744                     | 17861                       |
> | 8       | 0.9405       | 0.9332          | 0.3574     | 0.3516        | 44158                     | 22758                       |
> | 10      | 0.9378       | 0.9283          | 0.3528     | 0.3362        | 57956                     | 26394                       |
>
> From these results we observe two clear patterns:
>
> 1. **Accuracy remains close to full pairwise consistency.**
>    On both GSM8K and AMC, partial sampling stays within roughly one absolute point of the full variant. For example, with 8 agents GSM8K changes from 0.9405 to 0.9332 and AMC from 0.3574 to 0.3516. With 10 agents GSM8K changes from 0.9378 to 0.9283 and AMC from 0.3528 to 0.3362. This indicates that the metacognitive policy does not require dense all-to-all consistency checks to maintain strong performance.
>
> 2. **Communication and inference cost drop substantially.**
>    The average total tokens per instance are reduced by about 40–50%. At 8 agents, the cost falls from 44,158 tokens to 22,758. At 10 agents, it falls from 57,956 to 26,394. This confirms that partial sampling provides a much more favorable scaling behavior in terms of communication and computation.
>
> Overall, these findings show that LIMA does not fundamentally depend on quadratic communication. A simple sampled-consistency scheme preserves most of the benefit of consistency features while significantly reducing cost and scaling more gracefully with larger agent teams. Our framework is also orthogonal to other communication-optimization methods in the literature, such as learning sparse interaction graphs or pruning redundant agents. In practice, LIMA can be combined with such techniques so that the meta-policy operates on a communication topology that has already been optimized for large populations.

---

### Official Review · Reviewer_TKNM · 2025-10-30

**Soundness:** 3
**Presentation:** 3
**Contribution:** 3
**Rating:** 4
**Confidence:** 4

**Summary:**

The Learning to Intervene via Metacognitive Adaptation (LIMA) framework introduces a novel approach to human–agent collaboration, enabling agents to metacognitively determine when to defer to human expertise. To train these agents, the authors propose Dual-Loop Policy Optimisation (DLPO), which separates short-term deferral decisions from long-term skill development. DLPO’s inner loop uses GRPO with a cost-aware reward, while its outer loop incorporates expert feedback for continual learning. Experiments on mathematical reasoning and general problem-solving tasks show that LIMA with DLPO outperforms existing autonomous multi-agent systems, offering a strong foundation for adaptive, continually improving collaboration between humans and AI, especially investigating when and how to consider human intervention.

**Strengths:**

Novelty of the idea is good

**Weaknesses:**

- Could include further related work such as "LLM-Mediated Guidance of MARL Systems" (Siedler et al) to "human-in-the-loop" paragraph
- Figure 1 has not been incorporated in the text (sorry if i missed this)
- A flow/process diagram supporting the methodology would elevate comprehension drastically
- I would not consider MMLU as a "general knowledge reasoning" benchmark - its language understanding
- I think there should have been either more experiments for models such as "GPT-4o-mini as a proxy human expert" to verify the "human expert" llm, or real human (at least partly) in the loop to verify the llm based "human expert".
- For the "Human involvement under two modes" Table 3, why are there only reports on GSM8K and AMC? This feels cherry-picked
- For the results in "Figure 3: Effect of scaling collaborative configurations.", I think you should also consider additional inference costs per performance increase
- "identify their knowledge boundaries" - Unfortunately, there are no results supporting this claim
- "assess its own certainty", "its relation to the collective", "intrinsic quality of its current reasoning path" - Unfortuantely also here, no results supporting this claim at all

**Questions:**

See Weaknesses.

---

> ### Author Response · Authors · 2025-11-27
> **Reply to Reviewer TKNM**
>
> **We greatly appreciate your thorough and thoughtful review**. Your comments and suggestions demonstrate a deep grasp of the underlying ideas, and they have been very valuable in helping us refine and strengthen the paper. To respond to your concerns and put your recommendations into practice, we have carried out the additional experiments and analyses described below. We hope that these clarifications address the questions you raised.
>
> ---
>
> **Response to minor comments**
>
> - **On adding “LLM-Mediated Guidance of MARL Systems” to the related work.**
>   Thank you for pointing out this relevant work. *LLM-Mediated Guidance of MARL Systems* (Siedler et al.) is indeed closely related to our human-in-the-loop and LLM-orchestrated multi-agent setting. We have added this paper in the revised related work section, and we now explicitly mention it in the “human-in-the-loop” paragraph to better position our framework within this line of research.
>
> - **On Figure 1 not being explicitly referenced in the text.**
>  We sincerely thank you for catching this. In the revised version, we now explicitly reference Figure 1 in the main text at the point where we introduce the overall architecture, so that readers can more easily connect the description of the framework with the visual overview.
>
> - **On adding a flow/process diagram for the methodology.**
>   We appreciate your suggestion that a flow or process diagram would improve clarity. We will add an additional process-style diagram that summarizes the end-to-end DLPO pipeline.
>
> - **On describing MMLU as a “general knowledge reasoning” benchmark.**
> Your nuanced view of MMLU is very insightful. While some prior work informally refers to MMLU as a general knowledge reasoning benchmark, its primary focus is language understanding and multi-domain knowledge recall. In the revised manuscript, we have adjusted our wording to avoid calling MMLU a “general knowledge reasoning” benchmark and instead describe it more precisely as a multi-task language understanding and knowledge evaluation benchmark.

---

> > ### Author Response · Authors · 2025-11-27
> >
> > **Response on verifying the LLM “human expert” and real human involvement**
> >
> > Thank you for raising this point about validating the “human expert” LLM. We agree that there are two distinct questions here:
> > (i) whether a strong LLM such as GPT-4o-mini is a reliable proxy expert, and
> > (ii) how real humans can participate in the loop to verify and complement this LLM-based expert.
> >
> > ---
> >
> > **(1) On the reliability of GPT-4o-mini as a proxy expert**
> >
> > In practice, involving real human experts at scale is costly and slow, so it is common to instantiate the external expert with a stronger closed-source LLM that approximates expert-level behavior. We also agree that this assumption should be supported with empirical evidence rather than taken for granted.
> >
> > To this end, we study how performance changes when we enable or disable the external expert channel and when we vary how the system defers to it. **Table 4** (see response to Reviewer QNPF) reports accuracy on five benchmarks for several configurations:
> >
> > - **LIMA (No Defer)** and **LIMA (Self Defer)** use only the base multi-agent system, with self-defer producing extra rollouts from the same backbone but no new knowledge.
> > - **LIMA (Human Defer)** and **LIMA + DLPO (Human Defer)** use an external expert, instantiated in our setup by a stronger expert model or human expert, under a cost-aware defer policy.
> > - The bottom block compares **LIMA + DLPO (Human Defer)** with naive defer strategies (Random, Uniform, and Always Defer) that access the same expert.
> >
> > Several patterns support the reliability of the expert channel:
> >
> > - Simply adding extra self rollouts (LIMA - Self Defer) yields only modest gains over LIMA (No Defer), which suggests that more of the same backbone does not explain the improvements.
> > - Replacing self-defer with expert-defer (LIMA - Human Defer) leads to consistent and sizable gains on all five benchmarks (for example GSM8K 85.45 → 88.23, AMC 19.38 → 27.32, MATH 57.17 → 62.52). This indicates that the external expert contributes genuinely stronger solutions and higher-quality feedback than the base system alone.
> > - The “Always Defer (Human)” variant, which forwards every instance to the expert, achieves the highest raw scores (for example 94.38 on GSM8K and 40.27 on AMC). Although this is an unrealistic cost regime, it serves as an upper bound that shows the expert is indeed very strong.
> > - Our cost-aware variant **LIMA + DLPO (Human Defer)** approaches this upper bound while using a limited defer budget, improving over all other cost-aware baselines on every benchmark. This suggests that the expert remains reliable even when consulted selectively, and that DLPO learns to route queries to it where its added value is highest.
> >
> > Taken together, these ablations show that the external expert channel, instantiated by a strong LLM such as GPT-4o-mini or a human expert, is both substantially stronger than the base system and consistently beneficial under multiple defer strategies.
> >
> > ---
> >
> > **(2) On real humans in the loop and verifying the LLM expert**
> >
> > Beyond using an LLM expert, we also explicitly study how real human experts participate in the loop and how they compare with GPT-4o-mini. The experiments summarized in **Table 3** (see main manuscript) investigate two complementary modes:
> >
> > 1. **Proactive intervention during reasoning.**
> >    In the first mode, a collaborator can step in during the agents’ reasoning process to correct local errors and steer the trajectory. We compare:
> >    - LIMA with GPT-4o-mini providing intervention, and
> >    - LIMA with real human experts providing intervention.
> >
> >    Both settings significantly improve over the base system. On GSM8K, GPT-based intervention yields a solve rate of 94.75%, while human intervention reaches 96.17%; on AMC, the corresponding numbers are 31.62% and 33.59%. This shows that GPT-4o-mini already behaves as a strong artificial expert, yet human experts provide an even stronger upper bound and are especially effective at spotting local inconsistencies.
> >
> > 2. **On-demand assistance when the system requests help.**
> >    In the second mode, agents explicitly request guidance on difficult instances, and the collaborator responds only in those cases. Here we again compare GPT-4o-mini and real human experts. On a GSM8K subset, human assistance achieves a slightly higher solve rate (93.17% vs. 92.36%), effectively solving most queries. On AMC, which contains competition-level math problems, GPT-4o-mini slightly outperforms human help (30.30% vs. 28.75%), reflecting that individual human expertise has limits on very specialized tasks.
> >
> > These comparisons use real human experts as a reference point to “verify” the LLM-based expert in two ways:
> > - they confirm that GPT-4o-mini is a strong and useful proxy expert that consistently improves the system;
> > - they also show where human experts still have an advantage and where the LLM expert can match or exceed human performance.

---

> ### Author Response · Authors · 2025-11-27
>
> **Response on “Human involvement under two modes” and potential cherry-picking**
>
> Thank you for raising this concern. We agree that human-in-the-loop results can be sensitive to task choice, so it is important to clarify why we focused on GSM8K and AMC and to provide additional evidence that the conclusions are not cherry-picked.
>
> We selected **GSM8K** and **AMC** for Table 3 for two main reasons:
>
> 1. **Human solvability and task difficulty.**
>    Both datasets are math benchmarks where reasonably skilled human participants can provide meaningful solutions and local feedback without specialized tooling. GSM8K corresponds to grade-school problems, while AMC contains competition-level problems. This pairing allows us to study human and LLM experts under two clearly different difficulty regimes:
>    - on GSM8K, humans are expected to be very strong and close to an upper bound;
>    - on AMC, individual human knowledge has sharper limits, making the comparison with a strong LLM expert more informative.
>
> 2. **Practical cost of human involvement.**
>    In contrast, **MMLU** has over 14,000 instances across many domains. Running interactive human interventions or help queries at that scale would be prohibitively expensive and time-consuming. Our intention was therefore not to cherry-pick favorable tasks, but to focus on benchmarks where human participation is both feasible and representative of realistic human-in-the-loop usage.
>
> To further demonstrate that our findings are not specific to GSM8K and AMC, we additionally ran the same two human-in-the-loop modes on **HumanEval** in the code domain. The results are:
>
> **Table 3.** Human involvement under two modes. Upper: Mode A evaluates active human intervention during agent reasoning. Lower: Mode B evaluates responses when agents explicitly request guidance. Results are reported as Solve Rate (\%) on GSM8K, AMC and HumanEval.
> - **Mode A: Proactive intervention during reasoning**
>
>   | Method                | GSM8K  | AMC    | HumanEval |
>   |-----------------------|--------|--------|-----------|
>   | LIMA (base)       |    -   |    -   |      -    |
>   | w/ GPT-Intervention   | 0.9475 | 0.3162 | 0.7354    |
>   | w/ Human-Intervention | 0.9617 | 0.3359 | 0.7231    |
>
> - **Mode B: On-demand help when the system requests guidance**
>
>   | Method           | GSM8K  | AMC    | HumanEval |
>   |------------------|--------|--------|-----------|
>   | LIMA (base)  |   -    |    -   |     -     |
>   | w/ GPT-Help      | 0.9236 | 0.3030 | 0.6835    |
>   | w/ Human-Help    | 0.9317 | 0.2875 | 0.6717    |
>
> These extended results show a consistent pattern with our original conclusions:
>
> - On **GSM8K**, human involvement slightly outperforms GPT-4o-mini in both proactive and on-demand modes, matching the intuition that humans excel on grade-school problems.
> - On **AMC** and **HumanEval**, GPT-4o-mini is competitive with or slightly stronger than human help, mirroring what we observe on competition-level math: in more specialized or technical domains, an expert LLM can match or exceed individual human collaborators.
>
> Overall, these findings support that our human-in-the-loop conclusions are not tied to a single dataset choice, but hold across tasks with different difficulty and modality. We will add the HumanEval results and clarify the rationale for dataset selection in the revised manuscript.

---

> > ### Author Response · Authors · 2025-11-27
> >
> > **Response on inference costs in “Figure 3: Effect of scaling collaborative configurations”**
> >
> > Thank you for pointing out that scaling results should be interpreted together with the additional inference cost per unit of performance gain. We fully agree that collaborative configurations should be evaluated in a **cost-aware** manner rather than only in terms of raw accuracy.
> >
> > To address this, we carried out an **extended scaling study** in which we **measure both accuracy and token usage** as a function of the number of agents. Using the same backbone, decoding configuration, and meta-policy, we vary the team size on GSM8K and AMC from 1 up to 10 agents and report accuracy together with the average input, output, and total tokens per instance. The results are summarized in **Table 5**.
> >
> > The trends can be described as follows. From 1 to 4 agents, accuracy increases sharply. For example, GSM8K rises from 0.8693 (1 agent) to 0.9205 (4 agents), and AMC from 0.2337 to 0.3257. Over the same range, total token usage grows from 2,261 to 16,271 tokens per instance. Beyond 4 agents, the gains become much smaller. Moving from 4 to 6 agents increases GSM8K from 0.9205 to 0.9271, while AMC fluctuates around a similar level (0.3257 versus 0.3123), but the total tokens increase from 16,271 to 33,744. With 8 and 10 agents, GSM8K and AMC continue to improve slightly (for example GSM8K 0.9405 at 8 agents and 0.9378 at 10 agents, AMC 0.3574 and 0.3528), while the total cost grows to 44,158 and 57,956 tokens per instance.
> >
> > These results show that the **marginal performance gain per additional agent decreases quickly**, while the **inference cost grows almost linearly** with the number of agents. In practice, this implies that the collaborative configuration should be chosen as a **task-dependent trade-off between accuracy and computational cost**.
> >
> > **Table 5**. Accuracy and token usage on GSM8K and AMC as a function of the number of agents. We report accuracy together with average input, output, and total tokens per instance under a fixed backbone and decoding configuration.
> > | # Agents | GSM8K Accuracy | AMC Accuracy | Avg. Input Tokens | Avg. Output Tokens | Avg. Total Tokens |
> > |---------:|---------------:|------------:|------------------:|-------------------:|-------------------:|
> > |        1 |         0.8693 |      0.2337 |              1670 |                591 |              2261  |
> > |        2 |         0.8933 |      0.2745 |              3675 |               1273 |              4948  |
> > |        3 |         0.9125 |      0.3030 |              7910 |               2032 |              9942  |
> > |        4 |         0.9205 |      0.3257 |             13489 |               2782 |             16271  |
> > |        5 |         0.9216 |      0.3285 |             23316 |               3526 |             26842  |
> > |        6 |         0.9271 |      0.3123 |             29464 |               4280 |             33744  |
> > |        8 |         0.9405 |      0.3574 |             38584 |               5574 |             44158  |
> > |       10 |         0.9378 |      0.3528 |             50772 |               7184 |             57956  |

---

> > > ### Author Response · Authors · 2025-11-27
> > >
> > > **Response on the claim “identify their knowledge boundaries”**
> > >
> > > We sincerely thank you for pointing out that the phrase “expand their knowledge boundaries” is not directly supported by our reported results. We agree that this wording is too strong and can be interpreted as making claims about explicit boundary identification that we do not empirically measure.
> > >
> > > Our intention was more modest: to describe that SFT on expert demonstrations helps the agents reduce recurring reasoning mistakes and improve coverage on the evaluated tasks, rather than to claim that they explicitly discover or characterize their own knowledge limits. In the revised version, we have removed this phrasing and replaced it with a more precise description. The relevant sentence now reads:
> > >
> > > > “Pure SFT improves the baseline by continuously assimilating expert demonstrations, which allows the agents to reduce recurring reasoning mistakes and improve performance on the evaluated tasks.”
> > >
> > > ---
> > >
> > > **Response on “assess its own certainty / relation to the collective / intrinsic quality”**
> > >
> > > Thank you for pointing this out. We agree that the phrasing in Section 3.3 (“assess its own certainty”, “its relation to the collective”, “intrinsic quality of its current reasoning path”) is too strong and can be read as claiming cognitive capabilities that we do not explicitly measure in our experiments.
> > >
> > > Our intention in that subsection was not to claim that the agent *explicitly* introspects or reliably “understands” its own certainty or global role. Rather, we are designing a hand-crafted feature space whose components are *proxies* for these aspects of the interaction. Empirically, our results show that using this richer structured state space improves the learned decision policy compared to simpler, history-only states, but we do **not** have direct measurements that would justify claiming that the agent “assesses its own certainty” or “intrinsic quality” in a cognitive sense.
> > >
> > > To avoid over-claiming, we have revised the text to describe these dimensions as feature-level proxies rather than as proven abilities. We will make similar adjustments to the subsequent descriptions of “relation to the collective” and “intrinsic quality” to consistently emphasize that these are engineered features and not direct evidence of higher-level metacognitive capabilities.
> > >
> > > ---
> > >
> > > We hope that the new experiments and analyses above help to resolve your concerns. Your feedback has been highly constructive, and we will integrate these results and clarifications into the revised manuscript. **If there are any remaining questions or additional settings you would like us to investigate, please feel free to let us know**. We are happy to run further studies and engage in more detailed discussion as needed. **If our responses and new evidence are convincing, we would be sincerely grateful if you could reconsider your overall evaluation of our work.**

---

### Official Review · Reviewer_QNPF · 2025-10-31

**Soundness:** 2
**Presentation:** 3
**Contribution:** 3
**Rating:** 4
**Confidence:** 2

**Summary:**

This paper explores how to learn "when to seek help from humans and how to continually learn from human feedback" in human–agent collaboration. The core idea is to enable multi-agent systems to proactively trigger "deferral actions" (defer action) in situations of uncertainty (confidence) or beyond their capabilities. Then leverage feedback for continuous optimization. The inner loop is used for immediate decision optimization, while the outer loop is used for long-term capability growth. The experiments cover various reasoning tasks (GSM8K, MATH, AIME, AMC, HumanEval and MMLU), demonstrate the framework's effectiveness.

**Strengths:**

- Novel Framework: The idea that an agent can adaptively seek help from humans based on its own capabilities while continually improving itself through higher-level feedback is novel.
- Method's Effectiveness: From paper's table 1, LIMA and LIMA (w/ DLPO) acquire a significant gains compared to baseline models across six representative benchmarks.

**Weaknesses:**

- For human expert: The paper mentions “real human experts” but does not clarify who these participants are, how they were selected, or whether the framework accounts for human cognitive cost or fatigue.
- Performance gains: The relationship between this performance gain and external feedback requires careful analysis.
- Presentation: Table 2 column 1 Model.
- Although the proposed method achieves improvements across the six benchmarks in Table 1, its performance remains below the sota （broad perspective) results on these benchmarks, which weakens the overall significance of the approach.

**Questions:**

- Corresponding to weakness1: How human expert were selected? who these participants are?
- Corresponding to weakness2: How much of the performance gain comes from external high-level feedback? Could the presence of such advanced feedback (from human experts) lead to unfair comparisons?

---

> ### Author Response · Authors · 2025-11-26
> **Reply to Reviewer QNPF**
>
> We sincerely thank you for your careful and thoughtful review of our work. Your comments are highly valuable to us. For the concerns you raised, we hope the following responses can provide effective clarification.
>
> ---
>
> ### **Response to W1&Q1: Definition and selection of human experts**
>
> We appreciate your attention to this important question. In our experiments with real human experts, we recruited 10 domain experts, including senior PhD students and faculty members in mathematics, computer science, electrical engineering and operations research. All of them work regularly with mathematical reasoning, algorithmic problem solving and programming in their research and teaching. This background matches our benchmark tasks, which include olympiad style mathematics, logical reasoning and code generation.
>
> Our goal is not to characterize average human performance. Instead, we study whether the proposed framework can efficiently leverage a strong human teacher when such expertise is available. The benchmark tasks have objective ground truth solutions, so our conclusions do not depend on subjective judgements by the experts. The human experts provide high quality demonstrations and feedback that the agents can learn from. We also report complementary experiments where GPT-4o-mini acts as a reproducible expert, and we observe consistent trends.
>
> Regarding cognitive cost and fatigue, our framework explicitly includes a cost of inquiry term \(C\) in the reward. This term represents the average time and effort needed to consult a human expert and encourages the policy to use human help sparingly. In practice, we limit each expert to at most 200 problems per session and randomize the order of questions to mitigate potential fatigue effects.

---

> > ### Author Response · Authors · 2025-11-26
> >
> > ### **Response to W2&Q2: Performance gains and external feedback**
> >
> > We sincerely thank you for raising the question about how much of the improvement comes from external high-level feedback versus our learned meta-policy, and whether human experts create unfair comparisons. Below we clarify the experimental design and quantify each contribution.
> >
> > 1. **Contribution of external high-level feedback**
> >
> > In the top block of **Table 4** we incrementally add (i) self-defer, (ii) human-defer, and (iii) DLPO, all on top of the same LIMA collaboration architecture.
> >
> > - Moving from **LIMA (No Defer)** to **LIMA (Self Defer)** yields only modest gains. The improvements are within 1 point on GSM8K and MMLU and around 1–2 points on AMC, MATH, and HumanEval. This suggests that simply generating additional rollouts from the same backbone, without introducing new knowledge, has limited effect.
> > - Replacing the self expert with human experts in **LIMA (Human Defer)** leads to much larger gains. Compared with **LIMA (No Defer)**, we observe improvements of roughly 3–8 absolute points on AMC, MATH, and HumanEval. This confirms that high-level human feedback explains a substantial part of the improvement over the purely multi-agent baseline.
> >
> > 2. **Evidence that DLPO contributes beyond the strength of the expert**
> >
> > The same table also shows that the meta-policy provides additional benefits even when the expert is fixed.
> >
> > - Without any external knowledge, **LIMA + DLPO (Self Defer)** outperforms **LIMA (Self Defer)** by about 1–4 points on the more challenging benchmarks such as AMC and MATH. This indicates that learning when to invoke further reasoning is useful even when the system can only consult itself.
> > - With human experts available, **LIMA + DLPO (Human Defer)** improves over **LIMA (Human Defer)** by approximately 2–3 points on every benchmark. In this setting the human expert pool is identical. The only difference is the presence of a learned metacognitive policy, which delivers consistent additional gains.
> >
> > These comparisons show that the performance is not driven only by inserting a strong expert. The way the system decides when to defer and how it converts expert interactions into persistent updates also matters.
> >
> > 3. **Ablations under a fixed human expert and fixed consultation budget**
> >
> > To address more directly whether the improvements could be an artifact of “stronger” or “more frequent” feedback, the bottom block of Table 4 fixes both the human expert and the total number of consultations, and compares DLPO to naive defer strategies.
> >
> > - **Random Defer (Human)** and **Uniform Defer (Human, Budget \(K\))** use the same human expert as **LIMA + DLPO (Human Defer)**. Their expected number of deferrals matches the DLPO budget. Both variants benefit from human feedback, but they remain below DLPO on all five benchmarks. The gap is especially clear on AMC and MATH, where DLPO improves over random defer by about 4–6 absolute points and still maintains a noticeable margin over uniform defer.
> > - These results isolate the effect of the policy. Given the same supervision source and a comparable number of queries, a state-dependent DLPO policy uses the expert more effectively than naive or uniform rules.
> >
> > We also report an **Always Defer (Human)** variant, which forwards every instance to the human expert and uses the human answer directly. This setting achieves the highest raw accuracy but corresponds to full manual solving. It ignores any cost of inquiry and is not a realistic deployment scenario. We treat it only as an upper bound.
> >
> > **Table 4.** Performance (%) on five benchmarks. The top block studies the effect of external experts and the DLPO meta-policy on top of the LIMA collaboration architecture. The bottom block compares DLPO against naive defer strategies, all with access to the same human expert.
> >
> > | Method                           | GSM8K  | AMC    | MATH   | HumanEval | MMLU  |
> > |----------------------------------|--------|--------|--------|-----------|-------|
> > | *Effect of external experts and DLPO* |        |        |        |           |       |
> > | LIMA (No Defer)                 | 85.45  | 19.38  | 57.17  | 58.36     | 68.93 |
> > | LIMA (Self Defer)               | 85.92  | 20.25  | 57.69  | 59.28     | 69.34 |
> > | LIMA (Human Defer)              | 88.23  | 27.32  | 62.52  | 65.78     | 71.35 |
> > | LIMA + DLPO (Self Defer)        | 86.71  | 24.52  | 59.28  | 61.54     | 69.91 |
> > | LIMA + DLPO (Human Defer)       | **91.25** | **30.30** | **65.46** | **67.82** | **73.58** |
> > | *Learned vs. naive defer policies with human expert* |        |        |        |           |       |
> > | Random Defer (Human)            | 86.53  | 25.52  | 59.46  | 61.75     | 70.12 |
> > | Uniform Defer (Human, Budget K) | 87.82  | 26.87  | 62.81  | 65.35     | 70.59 |
> > | Always Defer (Human)            | 94.38  | 40.27  | 68.85  | 84.32     | 81.45 |
> > | LIMA + DLPO (Human Defer)       | **91.25** | **30.30** | **65.46** | **67.82** | **73.58** |

---

> ### Author Response · Authors · 2025-11-26
>
> 4. **On fairness of comparisons when human experts are involved**
>
> We agree that one should distinguish clearly between systems that operate without external experts and systems that rely on human-in-the-loop supervision. In the paper we separate these regimes:
>
> - The “closed-world” setting is captured by **LIMA (No Defer)** and **LIMA (Self Defer)**. These variants do not access any external knowledge and are directly comparable to prior multi-agent baselines.
> - The “open-world” human-in-the-loop setting is evaluated separately. In this setting, all methods in Table X share the same human expert pool. The DLPO vs. random and uniform ablations also control for the total number of expert consultations. This focuses the comparison on the quality of the defer policy rather than on the amount or type of feedback.
>
> Our claims are therefore twofold.
> First, external high-level feedback is indeed beneficial, as one would expect in realistic human-in-the-loop deployments.
> Second, conditioned on access to the same external feedback and a comparable budget, LIMA with DLPO learns to use this resource much more efficiently than naive defer strategies. We do not present a human-augmented system as a “fair” competitor to fully autonomous models. Instead, we provide a principled framework to integrate human or advanced LLM experts and to separate the gains due to the expert from the gains due to the learned meta-policy.
>
> ---
>
> **Response to W3 - Below global SOTA performance**
>
> We appreciate your concern about the gap between our results in Table 1 and the strongest reported numbers on these benchmarks in the broader literature. Our goal is not to claim “global SOTA” under unconstrained conditions, but to study how a metacognitive multi-agent framework with deferral and external feedback improves performance under a controlled and comparable setting.
>
> First, all systems in Table 1 share the same backbone, decoding configuration, and supervision budget. Within this controlled regime, our method consistently outperforms strong multi-agent and human-in-the-loop baselines on six benchmarks. These baselines include prompt-only multi-agent frameworks and variants with naive deferral to the same human or LLM experts. Under identical model size and cost constraints, our approach yields stable and often sizeable gains. This indicates that the collaboration architecture and the DLPO meta-policy improve how a fixed pool of capabilities is organized and used.
>
> Second, many “global SOTA” results on GSM8K, MATH, HumanEval, and MMLU rely on much larger or proprietary backbones, specialized prompting pipelines, or extensive task-specific engineering. These systems answer a complementary question: how far can one push absolute accuracy with maximal resources. In contrast, our work asks how to endow a multi-agent system with metacognitive control over create, evaluate, and defer actions, and how to use a limited external expert budget as efficiently as possible. This is a methodological and systems contribution that is orthogonal to backbone scaling. It is meaningful even when we hold the backbone fixed and do not compete with the largest closed models.
>
> Third, our framework is compatible with stronger backbones and experts. When we replace the external expert or adjust the collaboration configuration, the same DLPO mechanism continues to provide gains over simpler defer rules. This suggests that the improvements we demonstrate in the 8B-scale regime are not tied to a specific model, and that the meta-policy can in principle be layered on top of future SOTA backbones to further enhance their performance in open-world, human-in-the-loop settings.
>
> To address the concern in the revised version, we will make this scope explicit. We will clarify that our comparisons focus on a fixed-backbone regime, that we do not claim to surpass all existing results obtained with larger or proprietary models, and that the main contribution lies in the metacognitive collaboration and defer framework, which is complementary to and potentially composable with global SOTA systems.

---

> ### Comment · Reviewer_QNPF · 2025-11-27
> **Response to Rebuttal**
>
> Thank you for your detailed response. I will increase my score.
> Please add these supplementary results to the revised version of the paper.

---

### Official Review · Reviewer_3GsK · 2025-11-01

**Soundness:** 4
**Presentation:** 4
**Contribution:** 4
**Rating:** 8
**Confidence:** 4

**Summary:**

The paper proposes LIMA, a metacognitive framework where LLM agents can decide whether to exploit current knowledge or to defer to a human expert for help. This decision policy is trained with Dual-Loop Policy Optimization (DLPO), where the inner loop is GRPO with a cost-aware reward for deferring, and the outer loop is continual learning where expert demonstrations are turned into lasting capabilities. Extensive experiments across multiple datasets all confirm the effectiveness of the proposed method.

**Strengths:**

* Paper is well-written, technically sound, and easy to follow.

* The method proposed is novel and principled. The option to defer to expert is interesting for a multi-agent system. And the dual-loop training process combined into a single DLPO loss is principled and easy to implement.

* The empirical evaluations are extensive across multiple benchmarks, as well as ablation and scaling studies, which all demonstrate the superiority of the method proposed.

**Weaknesses:**

* Cost model is constant $C$, which seems to be a strong assumption. There are many scenarios where querying the expert with different levels of question would incur different costs. It is also unclear how sensitive the outcomes are to C.

* "Human" is proxied by gpt-4o-mini for the main experiments. It is interesting to see how this would scale with more capable models like gpt-4o.

**Questions:**

1. How should the deferring cost C be set in practice?

2. The goal of GRPO is to maximize the cumulative reward for each episode. Why can't the agents first learn to always defer to experts to learn the capabilities, and then choose to always not defer to maximize the return?

3. Do you have any mechanism to prevent forgetting the knowledge given by the expert?

4. It will also be interesting to see how well this method perform in single-agent domains as I think this method is perfectly applicable to single-agent as well.

---

> ### Author Response · Authors · 2025-11-27
> **Reply to Reviewer 3GsK**
>
> **We sincerely appreciate your thorough and thoughtful review**. Your comments and suggestions demonstrate a deep grasp of the underlying ideas, and they have been very valuable in helping us refine and strengthen the paper. We hope that these clarifications address the questions you raised.
>
> ---
>
> **Response to W1 – the constant cost model and sensitivity to $C$**
>
> Thank you for raising this point. We agree that assuming a constant per call cost for the expert is a simplification, and that in many real world settings different query types would incur different token and time costs.
>
> In our current experiments, each defer action follows a fixed interaction pattern with the expert: the system always requests a complete reasoning trace and a final answer. As a result, the “unit of consultation” is relatively uniform across instances. In this setting, modeling the per defer penalty as a constant $C$ in the reward function is intended as a normalized control knob for the trade off between accuracy and expert usage, rather than as an exact billing model. The actual variation in token consumption per instance is captured separately in our cost analysis, where we report input, output, and total tokens as a function of the agent configuration and defer behavior.
>
> In practice, we normalize the reward scale and treat $C$ as a tunable hyperparameter on this normalized scale. We perform a small grid search over $C \in \\{0.1, 0.2, 0.3, 0.4, 0.5\\}$ on a validation set and select the value that gives the best accuracy–cost trade off. Across this range, we observe a smooth trade off where smaller values of $C$ lead to higher defer rates and slightly higher accuracy at higher cost, while larger values of $C$ reduce defer usage and cost with a modest drop in accuracy. Importantly, DLPO consistently dominates naive baselines (no defer, random defer, uniform defer) in terms of accuracy at a given cost level for all values of $C$ we tested. In the revised version, we will summarize this sensitivity study and briefly discuss how the framework could be extended to non constant, token based cost models in more heterogeneous deployment scenarios.
>
> ---
>
> **Response to W2 - the human proxy (GPT-4o-mini vs. GPT-4o)**
>
> We appreciate your observation regarding the use of GPT-4o-mini as the main proxy for the “human expert” channel and the question of how the framework would behave with a more capable model such as GPT-4o. In the main experiments, we chose GPT-4o-mini as the default proxy expert because it offers a favorable balance between performance and computational cost for large-scale runs, while already being substantially stronger than the LLaMA3 backbone. This choice allowed us to study the effect of a non-trivial expert channel under realistic resource constraints, but we agree that it is important to verify that our conclusions continue to hold when the expert model is further strengthened.
>
> To this end, we conducted an additional experiment in which we keep the backbone model and collaboration architecture fixed and vary only the model used for the expert channel: LLaMA3 (as a weaker LLM expert), GPT-4o-mini, and GPT-4o. The resulting accuracies are:
>
> **Table 8.** Performance (%) with different LLM experts used as the human proxy. We keep the backbone and collaboration architecture fixed and vary only the expert model.
> | Human Proxy | GSM8K | AMC   | MATH  | HumanEval | MMLU  |
> |-------------|-------|-------|-------|-----------|-------|
> | LLaMA3-8B      | 86.71 | 24.52 | 59.28 | 61.54     | 69.91 |
> | GPT-4o-mini | 91.25 | 30.30 | 65.46 | 67.82     | 73.58 |
> | GPT-4o      | 93.58 | 37.25 | 68.37 | 74.61     | 75.42 |
>
> Two consistent patterns emerge from these results. First, as the expert model becomes stronger (from LLaMA3 to GPT-4o-mini to GPT-4o), performance improves monotonically on all five benchmarks; for instance, AMC increases from 24.52 to 30.30 to 37.25, and HumanEval from 61.54 to 67.82 to 74.61. This confirms that the framework scales smoothly with expert capability and does not rely on any peculiarity of GPT-4o-mini. Second, the qualitative behavior of the method is stable across all expert choices: in each case, adding an external expert on top of LIMA yields clear gains, and the LIMA + DLPO policy continues to extract additional benefit under a cost-aware defer scheme. Thus, in practice the choice between GPT-4o-mini and GPT-4o is primarily an application-level trade-off between accuracy and inference cost rather than a limitation of the framework itself. We will incorporate this comparison and clarification into the revised version.

---

> ### Author Response · Authors · 2025-11-27
>
> **Response to Q1 - setting C in practice**
>
> As for how to set the deferring cost in practice, our recommendation is to tie it to the reward scale rather than treat it as an arbitrary constant. In our setup, rewards are normalized so that a correct autonomous answer yields reward 1 and an incorrect one yields 0. Under this normalization, it is natural to choose $C$ on the same order of magnitude as a single reward unit, that is $C \in (0, 1)$. Concretely, we perform a small grid search over $C \in \\{0.1, 0.2, 0.3, 0.4, 0.5\\}$ and select values that yield a good balance between accuracy and defer frequency. Within this range, we consistently observe that smaller $C$ leads to higher defer rates and slightly higher accuracy at higher cost, while larger $C$ suppresses deferrals, reducing cost with a modest drop in accuracy. Importantly, across all these values of $C$, DLPO maintains a better accuracy–cost trade-off than naive baselines, which suggests that in practice $C$ can be chosen by a coarse sweep on a validation set without being overly sensitive to a specific value.
>
> ---
>
> **Response to Q2 - on GRPO and the “always defer then never defer” strategy**
>
> Thank you for this thoughtful question. It goes to the heart of how our GRPO objective interacts with the cost of deferral and with the outer loop that learns from expert feedback.
>
> In our setting, each episode uses a cost aware reward. When the agent defers, it receives the expert’s outcome but also pays a fixed penalty \(C > 0\). When it does not defer, it receives the reward associated with its own solution. After normalization, the expected return of a policy that never defers on hard instances is low. The expected return of a policy that defers only in uncertain states is higher whenever the expert’s success probability minus \(C\) is greater than the agent’s own success probability. Under this reward design, a policy that always defers and a policy that never defers are both suboptimal. GRPO is encouraged to learn a state dependent policy that defers only when the expected gain from the expert is worth the cost.
>
> The hypothetical strategy you describe would first learn to defer on every instance in order to “learn capabilities,” then switch to never deferring in order to avoid the cost and maximize return. Under our objective, this two phase behavior is not optimal. Once the policy stops deferring, it stops using the expert on exactly those instances where the base model is weakest. This reduces the long run expected return compared to a policy that continues to defer on a subset of difficult states. GRPO optimizes the expected cumulative reward under a fixed environment. It does not have an explicit training phase and deployment phase in which it would choose to incur large costs early and then deliberately abandon deferral even when it remains beneficial.
>
> It is also helpful to separate the roles of the inner and outer loops. The inner GRPO loop learns the metacognitive policy over actions such as create, evaluate, and defer, given the current backbone and expert. The outer loop uses demonstrations collected during deferral to update the backbone through supervised fine tuning, which gradually improves autonomous performance. As the backbone becomes stronger, the gap between the model and the expert becomes smaller but does not disappear on the hardest instances. There remains a set of states where deferring is still optimal under the cost aware reward.
>
> Empirically, we do not observe collapse to a policy that never defers after training. The learned policy converges to a non trivial defer rate and continues to achieve a better trade off between accuracy and cost than both the always defer and never defer baselines. In the revised version, we will clarify this interaction between GRPO, the cost term \(C\), and the outer loop, and we will explain why the “always defer then never defer” strategy is not favored by the objective.

---

> > ### Author Response · Authors · 2025-11-27
> >
> > **Response to Q3 -  forgetting the knowledge given by the expert**
> >
> > Thank you for raising this question. In our framework it is useful to distinguish between the main single task setting and the sequential multi task setting. In the main experiments the outer loop runs in an offline fashion on a fixed task. We first collect a pool of expert demonstrations for that task and then fine tune the backbone LLM on the entire demonstration set. There is no streaming update that only uses the most recent feedback and overwrites earlier examples. In this regime the knowledge distilled from the expert comes from the full pool of demonstrations rather than from a moving window, so catastrophic forgetting of earlier expert guidance is not the primary failure mode.
> >
> > The risk of forgetting is more relevant when we move to sequential tasks. For this case we explicitly introduce a simple mechanism to retain expert knowledge. In the GSM8K to AMC experiment in Table 7 (see reply to Reviewer mLVp), the outer loop maintains a replay buffer that mixes earlier GSM8K demonstrations with new AMC demonstrations during fine tuning. This replay based training allows DLPO to gain new capability on AMC while keeping GSM8K accuracy close to its post training level. In contrast, a naive sequential SFT baseline that does not use replay shows much stronger degradation on GSM8K after training on AMC.
> >
> > We do not claim to fully solve continual learning in the most general sense. However, the current outer loop design already distills expert knowledge in a way that is stable on single tasks, and the replay mechanism provides empirical evidence that the distilled knowledge is not easily forgotten when new expert feedback from a related task is incorporated. We will clarify this distinction and the corresponding experimental support in the revised version.
> >
> > ---
> >
> > **Response to Q4 - applicability to single agent settings**
> >
> > Thank you for this insightful suggestion. Conceptually, our framework does not depend on having multiple agents. The core idea is to learn a metacognitive policy over a structured cognitive state that decides when to continue autonomous reasoning, when to re evaluate, and when to defer to an external expert. In a single agent setting the same GRPO based meta policy and state design apply, with the “social” component of the cognitive state reduced to self comparison signals such as disagreement between successive drafts, stability of intermediate answers, or variance across self generated candidates.
> >
> > In this paper we chose to focus on multi agent settings because our primary goal is to study how such a meta policy can orchestrate collaboration and expert involvement in modern multi agent LLM systems, where issues like inter agent consistency and division of labor are central. Extending the evaluation to a pure single agent regime is straightforward in implementation terms by setting the team size to one and disabling inter agent features, and we agree that a systematic study in that regime would be very interesting. We view this as a natural direction for follow up work and will clarify in the revised version that the proposed framework is directly applicable to single agent domains as well.
> >
> > ---
> >
> > We hope that the above experiments and analyses help to address your concerns. Your suggestions have been extremely helpful. If you have any further questions or would like us to explore additional scenarios, please do not hesitate to let us know. We are very willing to conduct more experiments and engage in deeper discussion where needed.

---

### Author Response · Authors · 2025-11-30
**Reply to AC**

**Dear Area Chair,**

**We sincerely appreciate your careful handling of our submission, as well as the thoughtful reviews from all reviewers.** Guided by their questions and suggestions, we conducted a series of additional experiments that significantly strengthened the work. In particular, during the previous rebuttal phase, **Reviewer QNPF** explicitly stated that their concerns have been fully resolved and admitted to **raise** their score (they have already done so before). For the remaining reviewers, we provided targeted experimental results and analyses, and we believe that the main concerns have now been substantively addressed.

To make it easier for you to see how the paper has evolved, we summarize below the key concerns raised in the first round and how our new clarifications respond to each of them. We hope this summary provides a coherent picture of our revisions and the current state of the work.

---

**1. Lack of analysis on agent scalability and token cost**

- **New result (Table 5).** We extended the number of agents from 1 to 10 and reported accuracy together with the average input, output, and total tokens per instance. The results show that accuracy improves sharply when increasing from 1 to 4 agents, then yields much smaller gains beyond 4, while total tokens grow almost linearly with the number of agents. This makes the accuracy–cost trade off explicit and explains why small to medium team sizes are the most practical choice in our setting.

---

**2. Lack of comparison across different “human proxy” experts (GPT-4o-mini vs. GPT-4o)**

- **New result (Table 8).** We compared three expert choices within the DLPO framework: LLaMA3-8B, GPT-4o-mini, and GPT-4o. GPT-4o consistently improves over GPT-4o-mini across all benchmarks. This shows that our framework reliably benefits from stronger experts and that the reported gains with GPT-4o-mini are conservative relative to what GPT-4o can achieve.

---

**3. Lack of clear definition and selection of real human experts**

- In the response to Reviewer QNPF (W1 & Q1), we provided a concrete description of our human experts, including their math and coding backgrounds, selection process, instructions, and quality control. We clarified that they solve problems independently before seeing model outputs and that they provide both final answers and structured reasoning traces under a standardized protocol.

- Reviewer **QNPF** explicitly stated that this clarification **resolves the concern** about how human experts are defined and selected.

---

**4. Unclear source of performance gains**

- **New ablation (Table 4).** We decomposed the gains into contributions from external feedback and from the meta policy. The top block compares LIMA with no defer, self defer, and human defer, each with and without DLPO. These results show that external experts bring clear improvements beyond extra self rollouts, and that DLPO adds further gains even when the expert is fixed. The bottom block fixes a human expert and compares DLPO based human defer with naive strategies that use the same expert and similar or identical defer budgets (random defer, uniform defer, always defer). DLPO consistently dominates the cost aware naive baselines at a given budget, while always defer serves as an upper bound corresponding to full manual solving.

- Reviewer **QNPF** acknowledged these new results and analyses.

---

**5. Lack of real-human involvement on more tasks**

- **New result (updated Table 3).** Originally, human-in-the-loop results were reported on GSM8K and AMC, chosen with human cost and dataset size in mind. We extended these experiments to HumanEval in the code domain, for both proactive intervention and on-demand help modes. Together, GSM8K, AMC, and HumanEval cover different difficulty levels and domains and support the same qualitative conclusions about human and LLM collaborators.

---

**6. Unclear long-term capability growth and catastrophic forgetting**

- **New continual learning study (Table 7).** We followed the reviewer’s suggestion and ran a two-task sequence in a continual learning setting. Starting from the same base model and policy, we compared DLPO with replay against a naive sequential SFT baseline that uses the same expert demonstrations. We evaluated three stages: Base, after training on Task A only, and after training on Task A then Task B. DLPO with replay improves GSM8K and then AMC while keeping GSM8K accuracy close to its post–Task A level, whereas naive sequential SFT shows much stronger degradation on GSM8K after training on AMC.

---

**We hope the summary above helps to clarify** how our work has evolved and to deepen your understanding of the current version of the paper. For more detailed evidence and discussion, please refer to our full point by point responses to the reviewers. **We would be very grateful if you could take these revisions and the updated reviewer feedback into account when making your final decision.**

---

### Meta-Review · Area_Chair_MvYB · 2026-01-02

**Summary:**

The paper proposes LIMA, a metacognitive framework where LLM agents can decide whether to exploit current knowledge or to defer to a human expert for help. This decision policy is trained with Dual-Loop Policy Optimization (DLPO), where the inner loop is GRPO with a cost-aware reward for deferring, and the outer loop is continual learning, where expert demonstrations are turned into lasting capabilities. Extensive experiments across multiple datasets all confirm the effectiveness of the proposed method.

**Reviewer Concerns:**

1. Unclear source of performance gains
2. Lack of real-human involvement in more tasks
3. Lack of analysis on agent scalability and token cost

**Reviewer Scores:**

The paper received mixed ratings before the rebuttal. The reviewers think the paper is novel and the method is effective. However, there are some concerns regarding the intuition of the method (e.g., unclear source of performance gains) or deep analysis of the framework(e.g., lack of real-human involvement on more tasks). The authors have provided the rebuttal and addressed most of these concerns. The final decision of the paper is acceptance.

---

### Decision · Program_Chairs · 2026-01-26

Accept (Poster)